# APPROXIMATE CONDITIONAL COVERAGE & CALIBRATION VIA NEURAL MODEL APPROXIMATIONS

## ABSTRACT

A minimal required desideratum for quantifying the uncertainty from a classification model as a prediction set is class-conditional singleton set calibration. That is, such sets should map to the output of well-calibrated selective classifiers, matching the observed frequencies of similar instances. Recent works proposing adaptive and localized conformal p-values for deep networks do not guarantee this behavior, nor do they achieve it empirically. Instead, we use the strong signals for prediction reliability from KNN-based approximations of Transformer networks to construct data-driven partitions for Mondrian Conformal Predictors, which are treated as weak selective classifiers that are then calibrated via a new Inductive Venn Predictor, the VENN-ADMIT Predictor. The resulting selective classifiers are well-calibrated, in a conservative but practically useful sense for a given threshold, unlike conformal sets. They are inherently robust to changes in the proportions of the data partitions, and straightforward conservative heuristics provide additional robustness to covariate shifts. We compare and contrast to the quantities produced by recent Conformal Predictors on several representative and challenging natural language processing classification tasks, including class-imbalanced and distribution-shifted settings.

## 1 INTRODUCTION

Uncertainty quantification is hard. The *problem of the reference class* (see, e.g., Vovk et al., 2005, p. 159) necessitates task-specific care in interpreting even well-calibrated probabilities that agree with the observed frequencies. It is made harder in practice with deep neural networks, for which the otherwise strong blackbox *point* predictions are typically not well-calibrated and can *unexpectedly under-perform over distribution shifts*. And it is harder still for classification, given that the promising distribution-free approach of split-conformal inference (Vovk et al., 2005; Papadopoulos et al., 2002), an assumption-light frequentist approach suitable when sample sizes are sufficiently large, produces a counterintuitive p-value quantity in the case of classification (cf., regression).

**Setting.** In a typical natural language processing (NLP) binary or multi-class classification task, we have access to a computationally expensive blackbox neural model, $F$; a training dataset, $\mathcal{D}_{\mathrm{tr}} = \{Z_i\}_{i=1}^I = \{(X_i, Y_i)\}_{i=1}^I$ of $|\mathcal{D}_{\mathrm{tr}}| = I$ instances paired with their corresponding ground-truth discrete labels, $Y_i \in \mathcal{Y} = \{1, \dots, C\}$; and a held-out labeled calibration dataset, $\mathcal{D}_{\mathrm{ca}} = \{Z_j\}_{j=I+1}^{N=I+J}$ of $|\mathcal{D}_{\mathrm{ca}}| = J$ instances. We are then given a new test instance, $X_{N+1}$, from an unlabeled test set, $\mathcal{D}_{\mathrm{te}}$. One approach to convey uncertainty in the predictions is to construct a prediction set, produced by some set-valued function $\hat{\mathcal{C}}(X_{N+1}) \in 2^C$, containing the true unseen label with a specified level $1 - \alpha \in (0, 1)$ on average. We consider two distinct interpretations: As coverage and as a conservatively coarsened calibrated probability (after conversion to selective classification), both from a frequentist perspective.

**Desiderata.** For such prediction sets to be of general interest for classification, we seek **class-conditional singleton set calibration** (CCS). We are willing to accept noise in other size stratifications, but the singleton sets, $|\hat{\mathcal{C}}| = 1$, must contain the true value with a proportion of $\geq 1 - \alpha$, at least on average per class. We further seek **singleton set sharpness**; that is, to maximize the number of singleton sets. We seek reasonable **robustness** to distribution shifts. Finally, we seek **informative** sets that avoid the trivial solution of full cardinality.

If we are willing to fully dispense with specificity in the non-singleton-set stratifications for tasks with $|\mathcal{Y}| > 2$, our desiderata can be achieved, in principle, with selective classifiers.

**Definition 1** (Classification with reject option). *A selective classifier, $g : \mathcal{X} \to \mathcal{Y} \cup \{\perp\}$, maps from the input to either a single class or the reject option (represented here with the falsum symbol).*

**Remark 1** (Prediction sets are selective classifications). *The output of any set-valued function $\hat{\mathcal{C}}(X_{N+1}) \in 2^C$ corresponds to that of a selective classifier: Map non-singleton sets, $|\hat{\mathcal{C}}(X_{N+1})| \neq 1$, to $\perp$. Map all singleton sets to the corresponding class in $\mathcal{Y}$.*

To date, the typical approach for constructing prediction sets is not via methods for calibrating probabilities, but rather in the hypothesis testing framework of Conformal Predictors, which carry a PAC-style $(\alpha, \delta)$-valid coverage guarantee. In the inductive (or "split") conformal formulation (Vovk, 2012; Papadopoulos et al., 2002, inter alia), the p-value corresponds to confidence that a new point is as or more conforming than a held-out set with known labels. More specifically, we require a measurable function $A : \mathcal{Z}^I \times \mathcal{Z} \to \mathbb{R}$, which measures the conformity between $z$ and other instances. For example, given the softmax output of a neural network for $x$, $\hat{\pi} \in \mathbb{R}^C$, with $\hat{\pi}^y$ as the output of the true class, $A((z_1, \ldots, z_I), (x, y)) := \hat{\pi}^y$ is a typical choice. We construct a p-value, $v^{\hat{y}}$, as follows: $v^{\hat{y}} := \frac{|\{j=I+1,\ldots,N| \ \tau_j \leq \tau_{N+1}\}|+1}{N+1}$, where $\tau_j := A((z_1, \ldots, z_I), z_j)$, $\forall z_j \in \mathcal{D}_{\text{ca}}$ and $\tau_{N+1} := A((z_1, \ldots, z_I), (x_{N+1}, \hat{y}_{N+1}))$, where we suppose the true label is $\hat{y}$. We then construct the prediction set: $\hat{C}(x_{N+1}) = \{\hat{y} : v^{\hat{y}} > \alpha\}$. This is accompanied by a finite-sample, distribution-free coverage guarantee, which we state informally here.[1]

**Theorem 1** (Marginal Coverage of Conformal Predictors (Vovk et al., 2005)). *Provided the points of $\mathcal{D}_{\text{ca}}$ and $\mathcal{D}_{\text{te}}$ are drawn exchangeably from the same distribution $P_{XY}$ (which need not be further specified), the following marginal guarantee holds for a given $\alpha$: $\mathbb{P}\left\{Y_{N+1} \in \hat{\mathcal{C}}(X_{N+1})\right\} \geq 1 - \alpha$.*

The distribution of split-conformal coverage is Beta distributed (Vovk, 2012), from which a PAC-style $(\alpha, \delta)$-validity guarantee can be obtained, and from which we can determine a suitable sample size to achieve this coverage in expectation. Unfortunately, this does not guarantee singleton set coverage (the hypothesis testing analogue of our CCS desideratum), a known, but under-appreciated, negative result that motivates the present work:

**Corollary 1.** *Conformal Predictors do not guarantee singleton set coverage.* If they did, it would imply a stronger than marginal coverage guarantee.

**Existing approaches.** Empirically, Conformal Predictors are weak selective classifiers, limiting their real-world utility. We show this problem is not resolved by re-weighting the empirical CDF near a test point (Guan, 2022), nor by applying separate per-class hypothesis tests, nor by APS conformal score functions (Romano et al., 2020), nor by adaptive regularization RAPS (Angelopoulos et al., 2021), and *occurs even on in-distribution data*.

**Solution.** In the present work, we demonstrate, with a focus on Transformer networks (Vaswani et al., 2017), first that a closer notion of approximate conditional coverage obtained via the stronger validity guarantees of *Mondrian* Conformal Predictors is not sufficient in itself to achieve our desired desiderata. Instead, we treat such Conformal Predictors as weak selective classifiers, which serve as the underlying learner to construct a taxonomy for a Venn Predictor (Vovk et al., 2003), a valid *multi-probability* calibrator. This is enabled by data-driven partitions determined by KNN (Devroye et al., 1996) approximations, which themselves encode strong signals for prediction reliability. The result is a principled, well-calibrated selective classifier, *with a sharpness suitable even for highly imbalanced, low-accuracy settings, and with at least modest robustness to covariate shifts*.

## 2 Mondrian Conformal Predictors and Venn Predictors

A stronger than marginal coverage guarantee can be obtained by Mondrian Conformal Predictors (Vovk et al., 2005), which guarantee coverage within partitions of the data, including conditioned on the labels. Such Predictors are not sufficient for obtaining our desired desiderata, but serve as a principled approach for constructing a Venn taxonomy with a desirable balance between specificity vs. generality (a.k.a., over-fitting vs. under-fitting), the classic problem of the reference class.

---

[1]We omit the randomness component, which is not practically relevant at the sample sizes considered here.

Both Mondrian Conformal Predictors and Venn Predictors are defined by the choice of a particular **taxonomy**. A taxonomy is a measurable function $E : \mathcal{Z}^I \times \mathcal{Z} \to \mathcal{E}$, where $\mathcal{E}$ is a measurable space. A $E((z_1, \ldots, z_I), z)$ is referred to as a category, and corresponds to a classification of $z$, as via an attribute or label. The p-value of a Mondrian Conformal Predictor is then determined similarly to Conformal Predictors, but with conditioning on the category: $v^{\hat{y}} := \frac{|\{j = I+1, \ldots, N|\ e_j = e_{N+1} \wedge \tau_j \leq \tau_{N+1}\}| + 1}{N+1}$, where $e_j := E((z_1, \ldots, z_I), z_j)$, $\forall\ z_j \in \mathcal{D}_{\mathrm{ca}}$ and $e_{N+1} := E((z_1, \ldots, z_I), (x_{N+1}, \hat{y}_{N+1}))$. We will refer to the resulting coverage as approximate conditional coverage, a middle ground between marginal coverage and conditional coverage, which is not possible in the distribution-free setting with a finite sample (Lei & Wasserman, 2014):

**Theorem 2** (Approximate Conditional Coverage of Mondrian Conformal Predictors (Vovk et al., 2005; Vovk, 2012)). *Provided the points of $\mathcal{D}_{\mathrm{ca}}$ and $\mathcal{D}_{\mathrm{te}}$ are exchangeable within their categories defined by taxonomy E (Mondrian-exchangeability), the following coverage guarantee holds for a given $\alpha$: $\mathbb{P}\left\{Y_{N+1} \in \hat{\mathcal{C}}(X_{N+1}) \mid E(\cdot, (z_{N+1}))\right\} \geq 1 - \alpha$.*

**Venn Predictors** dispense with p-values (and coverage) and instead seek validity via calibration. They are multi-probability calibrators, in that they produce not one probability, but multiple probabilities for a single class, a compromise which yields an otherwise quite strong theoretical guarantee. Venn Predictors have a simple, intuitive appeal: They amount to calculating the empirical probability among similar points to the test instance. The quirk to enable the theoretical guarantee is that this is done by including the test point itself, assigning each possible label; hence, the generation of multiple empirical probability distributions. Specifically, for $x_{N+1}$ we first determine its category, typically some classification from the underlying model.[2] We will use $\mathcal{T}$ to indicate all instances in the category, where we have added $(x_{N+1}, c)$ assuming the true label is $c$. We then calculate the empirical probability:

$$p_c(c') := \frac{|\{(x^*,\ y^*) \in \mathcal{T} :\ y^* = c'\}|}{|\mathcal{T}|}, \forall\ c' \in \mathcal{Y} \tag{1}$$

We repeat this assuming each label is the true label, in turn, equivariant with respect to the taxonomy (that is, without respect to the ordering of points in the category). Remarkably, one of the probabilities from a multi-probability Venn Predictor is guaranteed to be perfectly calibrated. For our purposes, it will be sufficient to show this for the binary case. For a random variable $O \in [0, 1]$, such as the probablistic output of a classifier, and a binary random variable $Y \in \{0, 1\}$, we will follow previous work (Vovk & Petej, 2014) in saying $O$ is perfectly calibrated if $\mathbb{E}(Y \mid O) = O$ a.s. The validity of the Venn Predictor is then:

**Theorem 3** (Venn Predictor Calibration Validity (Theorem 1 Vovk & Petej (2014) )). *Provided the points of $\mathcal{D}_{\mathrm{ca}}$ and $\mathcal{D}_{\mathrm{te}}$ are IID, among the two probabilities output by a Venn Predictor for $Y = 1$, $p_0(1)$ and $p_1(1)$, one is perfectly calibrated.*

## 3 TASKS: CLASSIFICATION WITH TRANSFORMERS FOR NLP

The taxonomies will be chosen based on the need to partition the high-dimensional input of NLP tasks without having explicit attributes known in advance. We first introduce general notation for sequence labeling and document classification tasks. Each instance consists of a document, $x = x_1, \ldots, x_t, \ldots, x_T$, of $T$ tokens: Here, either words or amino acids. In the case of **supervised sequence labeling (SSL)**, we seek to predict $\hat{y} = \hat{y}_1, \ldots, \hat{y}_t, \ldots, \hat{y}_T$, the token-level labels for each token in the document, and we have the ground-truth token labels, $y_t$, for training. For **document classification (DC)**, we seek to predict the document-level label $\hat{y}$, and we have $y$ at training.

For each task, our base model is a Transformer network. After training and/or fine-tuning, we fine-tune a kernel-width 1 CNN (MEMORY LAYER) over the output representations of the Transformer, producing predictions and representative dense vectors at a resolution (e.g., word-level or document-level) suitable for each task. Following past work, we will refer to these representations as "exemplar vectors" primarily to contrast with "prototype", which is sometimes taken to refer to class-centroids,

---

[2]Existing taxonomies for Venn Predictors include using the predictions from a classifier (Lambrou et al., 2015), variations on nearest neighbors (Johansson et al., 2018), and isotonic regression, the Venn-ABERS Predictor (Vovk & Petej, 2014; Vovk et al., 2015).

Table 1: Overview of experiments.

| Label | Task | $|\mathcal{Y}|$ | $|\mathcal{D}_{\text{valid}}|$ | $|\mathcal{D}_{\text{te}}|$ | $r$ | Base network | Acc. | Characteristics |
|---|---|---|---|---|---|---|---|---|
| PROTEIN | SSL | 3 | 560k | {30k,7k} | $\mathbb{R}^{1000}$ | $\sim$ BERT$_{\text{BASE}}$ | Mid | In-domain (2 test sets) |
| GRAMMAROOD | SSL | 2 | 35k | 93k | $\mathbb{R}^{1000}$ | BERT$_{\text{LARGE}}$ | Low | Domain-shifted+imbalanced |
| SENTIMENT | DC | 2 | 16k | 488 | $\mathbb{R}^{2000}$ | BERT$_{\text{LARGE}}$ | High | In-domain (acc. $> 1 - \alpha$) |
| SENTIMENTOOD | DC | 2 | 16k | 5k | $\mathbb{R}^{2000}$ | BERT$_{\text{LARGE}}$ | Mid-Low | Domain-shifted/OOD |

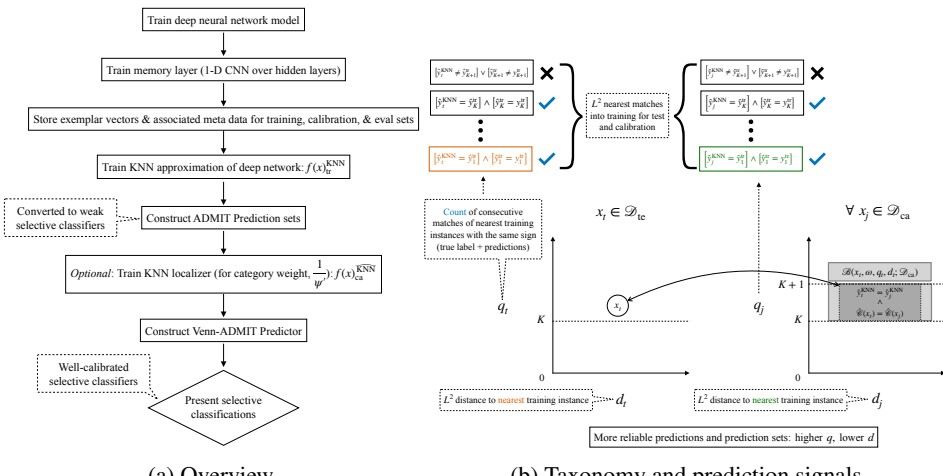

(a) Overview         (b) Taxonomy and prediction signals

Figure 1: On the *left* is a high-level overview. The *right* illustrates a category assignment for the VENN-ADMIT Predictor, and the key prediction signals enabled by $f(x)_{\text{tr}}^{\text{KNN}}$: Predictions become more reliable with increased label and prediction matches into training ($q$) and lower distances to training ($d$). The resulting well-calibrated selective classifiers are robust to changes in the proportion of these categories.

whereas the "exemplars" are unique to each instance. We will subsequently use $f(x_t) \in \mathbb{R}^C$ for the prediction logits produced by the MEMORY LAYER corresponding to the token at index $t$; $\pi^c$ as the corresponding softmax normalized output for class $c$; and $r_t$ as the associated exemplar vector. For **SSL** there are $T$ such logits and vectors. For **DC**, $t$ corresponds to a single representation of the document, with $f(x_t)$ formed by a combination of local and global predictions (as described further in the Appendix). In the present work, we will primarily only be concerned with Transformers at the level of abstraction of the exemplar vectors, $r_t$; we refer the reader to the original works describing Transformers (Vaswani et al., 2017) and the particular choice for the MEMORY LAYER (Schmaltz, 2021) for additional details. **Splits.** Our baselines of comparisons use $\mathcal{D}_{\text{tr}}$, $\mathcal{D}_{\text{ca}}$, as the training and calibration sets, respectively. For our methods, we will assume the existence of an additional disjoint split of the data, $\mathcal{D}_{\text{knn}}$ for setting the parameters of the KNNs. We will also require two calibration sets, $\mathcal{D}_{\text{ca}}^{\text{mc}}$, which serves as the calibration set for the Mondrian Conformal Predictor, and $\mathcal{D}_{\text{ca}}^{\text{vp}}$, which serves as the calibration set for the Venn Predictor.

## 4 METHODS

We first define the taxonomy for our Mondrian Conformal Predictor, ADMIT. The resulting sets will serve as baselines of comparisons, but will primarily be used as weak selective classifiers for defining the taxonomy of our Venn Predictor, the VENN-ADMIT Predictor. In both cases, we make use of non-parametric approximations of Transformers, which encode strong signals for prediction reliability, including over distribution-shifts, and are at least as effective as the model being approximated (Schmaltz, 2021). Predictions become less reliable at $L^2$ distances farther from the training set and with increased label and prediction mismatches among the nearest matches. We further introduce an additional KNN approximation that serves as a localizer, relating a test instance to the distribution of the conformal calibration set, which serves as a conservative heuristic for category assignments. Figure 1 provides an overview of the components and a visualization of the prediction signals and Venn taxonomy.

### 4.1 KNN APPROXIMATION OF A TRANSFORMER NETWORK

In order to partition the feature space, we first approximate the Transformer as a weighted combination of predictions and labels over $\mathcal{D}_{\text{tr}}$ (Section 4.1.1). This approximation, $f(x)_{\text{tr}}^{\text{KNN}}$, then becomes the model we use in practice, rather than the logits from the Transformer itself. We use this approximation to partition the data via a feature that separates more reliable points from less reliable points (Section 4.1.2) and via a distance-to-training band (Section 4.1.3).

#### 4.1.1 RECASTING A TRANSFORMER PREDICTION AS A WEIGHTING OVER THE TRAINING SET

We adapt the distance-weighted KNN approximation of Schmaltz (2021) for the multi-class setting. As in the original work, the KNN is trained to minimize prediction mis-matches against the output of the MEMORY LAYER (*not* the ground-truth labels). Training is performed on a 50/50 split of $\mathcal{D}_{\text{knn}}$, as described further in Appendix C:

$$f^c(x_t) \approx f^c(x_t)_{\text{tr}}^{\text{KNN}} = \beta^c + \sum_{\substack{k \in \underset{i \in \{1,\ldots,|\mathcal{D}_{\text{tr}}|\}}{\arg \text{K} \min} ||\boldsymbol{r}_t - \boldsymbol{r}_i||_2}} w_k \cdot \left( \tanh(f^c(x_k)) + \gamma^c \cdot \tilde{y}^c \right), \quad (2)$$

$$\text{where } w_k = \frac{\exp\left(-||\boldsymbol{r}_t - \boldsymbol{r}_k||_2/\eta\right)}{\sum_{\substack{k' \in \underset{i \in \{1,\ldots,|\mathcal{D}_{\text{tr}}|\}}{\arg \text{K} \min} ||\boldsymbol{r}_t - \boldsymbol{r}_i||_2}} \exp\left(-||\boldsymbol{r}_t - \boldsymbol{r}_{k'}||_2/\eta\right)} \quad (3)$$

$\tilde{y}^c$ is the ground-truth label ($y$ for **DC**, $y_t$ for **SSL**) for class $c$ transformed to be in $\{-1, 1\}$. $K$ is small in practice; $K = 25$ in all experiments here, and in general can be chosen using $\mathcal{D}_{\text{knn}}$. This approximation has $2 \cdot C + 1$ learnable parameters, corresponding to $\beta^c$ and $\gamma^c$ for each class, and the temperature parameter $\eta$. We indicate the softmax normalized output for each class with $\pi^c(x_t)_{\text{tr}}^{\text{KNN}}$. This model is used to produce approximations over all calibration and test instances.

#### 4.1.2 DATA-DRIVEN FEATURE-SPACE PARTITIONING: TRUE POSITIVE MATCHING CONSTRAINT

For each calibration and test point, we define the feature $q_t \in [0, K]$ as the count of *consecutive* sign matches of the prediction of the KNN, $\hat{y}_t^{\text{KNN}}$, with the true label and MEMORY LAYER prediction of the up to $K$ nearest matches from the *training* set, $\mathcal{D}_{\text{tr}}$:

$$q_t(K) = \sum_{\substack{k \in \underset{i \in \{1,\ldots,|\mathcal{D}_{\text{tr}}|\}}{\arg \text{K} \min} ||\boldsymbol{r}_t - \boldsymbol{r}_i||_2}} \left[ \hat{y}_t^{\text{KNN}} = \hat{y}_k^{\text{tr}} \right] \wedge \left[ \hat{y}_k^{\text{tr}} = y_k^{\text{tr}} \right] \wedge [q_t(k-1) = k-1], \quad (4)$$

with $q(0) := 0$. We further also use the $L^2$ distance to the nearest training set match as a basis for subsetting the distribution into distance bands, as discussed in the next section:

$$d_t = \min ||\boldsymbol{r}_t - \boldsymbol{r}_i||_2, i \in \{1, \ldots, |\mathcal{D}_{\text{tr}}|\} \quad (5)$$

#### 4.1.3 DATA-DRIVEN FEATURE-SPACE PARTITIONING: DISTANCE-TO-TRAINING BAND

We define the partition, $\mathcal{B}$, around each $x_t \in \mathcal{D}_{\text{te}}$, constrained to $q$, as the $L^2$-distance-to-training band with a radius of $\omega = \delta \cdot \hat{s}$, with $\delta \in \mathbb{R}^+$ as a user-specified parameter and $\hat{s}$ as the estimated standard deviation of constrained true positive calibration set distances, $\hat{s} = \text{std}([d_j : j \in \{I + 1, \ldots, I + |\mathcal{D}_{\text{ca}}|\}, q_j > 0, \hat{y}_j^{\text{KNN}} = y_j])$:

$$\mathcal{B}(x_t, \omega, q_t, d_t; \mathcal{D}_{\text{ca}}) = \{x_j : x_j \in \mathcal{D}_{\text{ca}}, d_j \in [d_t - \omega, d_t + \omega], q_t = q_j\} \quad (6)$$

### 4.2 PREDICTION SETS WITH APPROXIMATE CONDITIONAL COVERAGE: ADMIT

We then define a taxonomy for our Mondrian Conformal Predictor by the partitions defined by $\mathcal{B}$ *and* the true labels: $E(\cdot, (z_t)) = \mathcal{B}(x_t, \omega, q_t, d_t; \mathcal{D}_{\text{ca}}) \wedge y$. This conditioning on the labels means we apply split-conformal prediction separately for each label, "label-conditional" conformal prediction (Vovk et al., 2005; Vovk, 2012; Sadinle et al., 2018), which provides built-in robustness to label proportion shifts (c.f., Podkopaev & Ramdas, 2021). We will refer to this method and the resulting sets with the label ADMIT. We always include the predicted label in the set. Pseudo-code appears in Appendix E.

### 4.3 Inductive Venn-ADMIT Predictors & Selective Classifiers

An ADMIT Predictor maps to a weak selective classifier. We instead seek a well-calibrated selective classifier, which we define as follows, as a straightforward coarsening of the probability, only calculated over the admitted subset:

**Definition 2** (Well-calibrated selective classifiers). *We take as $S \in [0, 1]$ the random variable indicating the probability a non-rejected prediction from a selective classifier, g, should be admitted. We will say a selective classifier is conservatively well-calibrated (or just "well-calibrated") if $\mathbb{E}(Y \mid S \geq 1 - \alpha) \geq 1 - \alpha$ for a given $\alpha \in (0, 1)$.*

We construct such a selective classifier, $g$, as follows. Construct ADMIT sets for $\mathcal{D}_{\text{ca}}^{\text{vp}}$ and $\mathcal{D}_{\text{te}}$, in both cases using $\mathcal{D}_{\text{ca}}^{\text{mc}}$ as the calibration set. Next, convert the ADMIT sets into selective classifiers, $g_{\text{weak}}$, as in Remark 1. Calibrate the non-rejected predictions of $\mathcal{D}_{\text{te}}$ (i.e., the $\hat{y}^{\text{KNN}}$ predictions that were singleton sets and now admitted predictions of $g_{\text{weak}}$) using a Venn Predictor with a taxonomy defined by $\mathcal{B}$ and the prediction of the KNN, $\hat{y}^{\text{KNN}}$, now using $\mathcal{D}_{\text{ca}}^{\text{vp}}$ as the calibration set. The VENN-ADMIT selective classifier is then the following decision rule, where $p_0(1), p_1(1)$ are the two Venn probabilities associated with $g_{\text{weak}}$:

$$g(x_t) = \begin{cases} \perp & \text{if } \min(p_0(1), p_1(1)) < 1 - \alpha \\ \hat{y}_t^{\text{KNN}} & \text{otherwise} \end{cases} \tag{7}$$

We can then take as $S \coloneqq 1$, if $g(x_t) \neq \perp$ (as used in Def. 2); i.e., a coarsening of the probability of the points admitted by $g$.

**Proposition 1** (VENN-ADMIT selective classifiers are well-calibrated). *Provided the points of $\mathcal{D}_{\text{ca}}^{\text{vp}}$ and $\mathcal{D}_{\text{te}}$, restricted to $\mathcal{B}$, are IID, the selective classifier g defined by Eq. 7 is well-calibrated in the sense of Definition 2.*

This follows directly from Theorem 3. □

### 4.4 Robustness

VENN-ADMIT selective classifiers are robust to covariate shifts that correspond to changes in the proportion of the partitions. We propose two simple heuristics that provide additional robustness. We first state two useful propositions that will justify the heuristics.

**Proposition 2** (VENN-ADMIT calibration invariance to partition censoring). *VENN-ADMIT selective classifiers remain well-calibrated in the sense of Definition 2 with censoring of 1 or more partitions $\mathcal{B}$.*

This directly follows from the fact that both the weak selective classifier (ADMIT) and the well-calibrated selective classifier (VENN-ADMIT) treat each partition independently. We can thus construct a new selective classifier, $g'$, that maps any input in the censored partition(s) to $\perp$. □

**Proposition 3** (VENN-ADMIT calibration invariance to test point up-weighting). *VENN-ADMIT selective classifiers remain well-calibrated in the sense of Definition 2 using any test point weight $[1, \infty)$ when calculating the empirical probabilities of the VENN-ADMIT Predictor.*

Increasing the test point weight above 1 can only decrease the lower probability produced by the VENN-ADMIT Predictor (since the denominator in Eq. 1 can only increase). Since Def. 2 is only calculated for admitted points, this notion of conservative well-calibration is retained. □

#### 4.4.1 Censoring Less Reliable Data Partitions

The feature $q$ can be viewed as an ensemble across multiple similar instances from training. Greater agreement suggests greater confidence in the prediction. We can restrict to partitions with the maximum value, $q = K = 25$, here. By Prop. 2, calibration of the selective classifier is maintained.

#### 4.4.2 Localized up-weighting based on category similarity

We can up-weight the test point when calculating the VENN-ADMIT probabilities using an additional KNN localizer, $f(x)_{\text{ca}}^{\widehat{\text{KNN}}}$, in this case with $\mathcal{D}_{\text{ca}}$ as the support set of the KNN and a single

parameter, a temperature weight. Weights increase above 1 with greater dissimilarity between a test point and its assigned category. Additional details in Appendix B. By Prop. 3, calibration of the selective classifier is maintained.

### 4.4.3 Robust Venn-ADMIT Selective Classifications

For a given test point, $x_t$, we first construct a weak selective classifier with the ADMIT procedure of Section 4.2, followed by calibration via the Venn-ADMIT Predictor (Section 4.3). Optionally, we apply the heuristics described in Sections 4.4.1 and 4.4.2. Pseudo-code appears in Appendix E.

## 5 Experiments

We have established that the Venn-ADMIT selective classifications are conservatively well-calibrated; however, we have not said anything about the proportion of points that will be admitted. If the procedure is unnecessarily strict, we may nonetheless prefer the output from alternative approaches, such as Conformal Predictors. Additionally, the Venn-ADMIT Predictor is inherently robust to changes in the proportions of the data partitions, but whether that corresponds to real-world distribution shifts is task and data dependent. To address these concerns, we turn to empirical evaluations. We evaluate on a wide-range of representative NLP tasks, including challenging domain-shifted and class-imbalanced settings, and in settings in which the point prediction accuracies are quite high (marginally $> 1 - \alpha$) and in which they are relatively low. We follow past work in setting $\alpha = 0.1$ in our main experiments. We set $\delta = 1$. We summarize and label our benchmark **tasks**, the underlying parametric networks, and data in Table 1 A disjoint set of size 144k, the CB513 set from PROTEIN, was used for initial methods development. In the Table, $\mathcal{D}_{\text{valid}}$ is the original held-out validation set associated with each task. For the ADMIT and Venn-ADMIT approaches, a random 10% sample of $\mathcal{D}_{\text{valid}}$ serves as the disjoint $\mathcal{D}_{\text{knn}}$ set for training the KNNs, with the remaining data split evenly for $\mathcal{D}_{\text{ca}}^{\text{mc}}$ and $\mathcal{D}_{\text{ca}}^{\text{vp}}$. The baseline and comparison methods are given the full $\mathcal{D}_{\text{valid}}$ as $\mathcal{D}_{\text{ca}}$. The Appendix provides implementation details on constructing the exemplar vectors, $\boldsymbol{r}$, for each of the tasks from the MEMORY LAYER.

### 5.1 Comparison Models

As a distribution-free **baseline** of comparison we consider the size- and adaptiveness-optimized RAPS algorithm of Angelopoulos et al. (2021), $\text{RAPS}_{\text{SIZE}}$ and $\text{RAPS}_{\text{ADAPT}}$, which combine regularization and post-hoc Platt-scaling calibration (Platt, 1999; Guo et al., 2017), on the output of the MEMORY LAYER. Using stratification of coverage by cardinality as a metric, $\text{RAPS}_{\text{ADAPT}}$, in particular, was reported to more closely approximate conditional coverage than the alternative APS (Romano et al., 2020), with smaller sets. $\text{CONF}_{\text{BASE}}$ is a split-conformal point of reference for simply using the output of $f(x)_{\text{tr}}^{\text{KNN}}$ without further conditioning, nor post-hoc calibration. $\text{LOCAL}_{\text{CONF}}$ is a localized conformal (Guan, 2022) baseline using the KNN localizer $f(x)_{\text{ca}}^{\widehat{\text{KNN}}}$. Across methods, the point prediction is included in the set, which ensures conservative (but not necessarily exact/upper-bounded) coverage by eliminating null sets.

We use the label ADMIT to indicate the Mondrian Conformal sets. We use the label Venn-ADMIT to indicate Venn-ADMIT selective classifications with test-point up-weighting with the KNN localizer (Sec. 4.4.2), and $\text{Venn-ADMIT}_{qK}$ as those with the further restriction of $q = K$ (Sec. 4.4.1). The results Venn-ADMIT-w exclude test-point up-weighting; $\text{Venn-ADMIT}_{qK}$-w excludes test-point up-weighting, but restricts to $q = K$.

Calibration in general is difficult to evaluate, with conflicting definitions and metics (Kull et al., 2019; Gupta & Ramdas, 2022, inter alia). In the hypothesis testing framework, approaches have been proposed to make marginal Conformal Predictors more adaptive (i.e., to achieve closer approximations to conditional coverage), but evaluations omit class-wise singleton set coverage, arguably the baseline required quantity needed in practice for classification. In contrast, our desiderata are easily evaluated and resolve these concerns: Of the admitted points, we calculate the proportion of points matching the true label, $\overline{y \in \mathcal{C}}$, for each class. That is, given an admitted prediction (or similarly, a singleton set), an end-user should have confidence that the per-class accuracy is at least $1 - \alpha$. Additionally, other things being equal, the proportion of admitted points ($\frac{n}{N}$) should be maximized.

Table 2: Model approximation vs. MEMORY LAYER accuracy/$F_{0.5}$.

| Model/Approx. | PROTEIN (ACC.) | | | GRAMMAROOD ($F_{0.5}$) | | SENTIMENT (ACC.) | | SENTIMENTOOD (ACC.) | |
|---|---|---|---|---|---|---|---|---|---|
| | $\mathcal{D}_{\text{ca}}$ | TS115 | CASP12 | $\mathcal{D}_{\text{ca}}$ | $\mathcal{D}_{\text{te}}$ | $\mathcal{D}_{\text{ca}}$ | $\mathcal{D}_{\text{te}}$ | $\mathcal{D}_{\text{ca}}$ | $\mathcal{D}_{\text{te}}$ |
| MEMORY LAYER | 0.75 | 0.77 | 0.70 | 0.59 | 0.40 | 0.92 | 0.93 | 0.92 | 0.78 |
| $f(x)_{\text{tr}}^{\text{KNN}}$ | 0.76 | 0.77 | 0.71 | 0.58 | 0.43 | 0.92 | 0.93 | 0.92 | 0.79 |
| $f(x)_{\text{ca}}^{\widetilde{\text{KNN}}}$ | - | 0.77 | 0.70 | - | 0.42 | - | 0.93 | - | 0.78 |

Table 3: The empirical behavior of the calibration points differs significantly with $q = 0$ vs. $q = K$, and as the distance to training ($d_t$) varies, in terms of $f(x)_{\text{tr}}^{\text{KNN}}$ point accuracy (ACC.), and the distribution of over-confidence and under-confidence (reflected in $\hat{\tau}^{0.1}$, 0.1 quantile threshold). (Validation set of PROTEIN.)

| Subset | PROTEIN: Class Label (Amino-Acid/Token-Level Sequence Labeling) | | | | | | | | | | | |
|---|---|---|---|---|---|---|---|---|---|---|---|---|
| | $y = $ **HELIX** | | | $y = $ **STRAND** | | | $y = $ **OTHER** | | | $y \in \{\mathbf{H}, \mathbf{S}, \mathbf{O}\}$ | | |
| | $\hat{\tau}_c^{0.1}$ | ACC. | $\frac{n}{N}$ | $\hat{\tau}_c^{0.1}$ | ACC. | $\frac{n}{N}$ | $\hat{\tau}_c^{0.1}$ | ACC. | $\frac{n}{N}$ | $\hat{\tau}^{0.1}$ | ACC. | $\frac{n}{N}$ |
| $q = 0$ | 0.07 | 0.59 | 0.07 | 0.07 | 0.56 | 0.05 | 0.18 | 0.56 | 0.10 | 0.11 | 0.57 | 0.22 |
| $q = K$ | 0.96 | 0.98 | 0.12 | 0.95 | 0.94 | 0.04 | 0.92 | 0.92 | 0.08 | 0.94 | 0.96 | 0.24 |
| $q \in [0, K]$ | 0.12 | 0.81 | 0.37 | 0.06 | 0.70 | 0.21 | 0.13 | 0.74 | 0.42 | 0.11 | 0.76 | 1. |
| $d_t < $ median | | | | | | | | | | | | |
| $\quad q = 0$ | 0.09 | 0.64 | 0.02 | 0.07 | 0.65 | 0.01 | 0.16 | 0.55 | 0.03 | 0.12 | 0.60 | 0.07 |
| $\quad q = K$ | 0.96 | 0.98 | 0.07 | 0.95 | 0.96 | 0.03 | 0.93 | 0.94 | 0.06 | 0.94 | 0.96 | 0.15 |
| $\quad q \in [0, K]$ | 0.27 | 0.87 | 0.16 | 0.09 | 0.81 | 0.08 | 0.14 | 0.79 | 0.18 | 0.16 | 0.82 | 0.43 |
| $d_t \geq $ median | | | | | | | | | | | | |
| $\quad q = 0$ | 0.07 | 0.57 | 0.05 | 0.07 | 0.53 | 0.04 | 0.19 | 0.57 | 0.07 | 0.11 | 0.56 | 0.16 |
| $\quad q = K$ | 0.96 | 0.98 | 0.05 | 0.04 | 0.90 | 0.01 | 0.03 | 0.87 | 0.02 | 0.93 | 0.94 | 0.09 |
| $\quad q \in [0, K]$ | 0.09 | 0.76 | 0.21 | 0.05 | 0.62 | 0.12 | 0.13 | 0.70 | 0.24 | 0.09 | 0.70 | 0.57 |

## 6 RESULTS

Across tasks, the KNNs consistently achieve similar point accuracies as the base networks (Table 2). This justifies their use in replacing the output logit of the underlying Transformers. Table 3 then highlights our core motivations for leveraging the signals from the KNNs: There are stark differences across instances as $q$ increases and as the distance to training increases (shown here for PROTEIN, but observed across tasks). In order to obtain calibration, coverage, or even similar point accuracies, on datasets with proportionally more points with $q < K$, and/or far from training, we must control for changes in the proportions of these partitions.

Table 4 and Table 5 contain the results. The Conformal Predictors RAPS and APS behave as advertised, obtaining marginal coverage (not shown) for in-distribution data. However, the additional adaptiveness of these approaches does not translate into reliable singleton set coverage. More specifically: **The Conformal Predictors tend to only obtain singleton set coverage when the point accuracy of the model is $\geq 1 - \alpha$, including over in-distribution data.** Only for the high-accuracy, in-distribution SENTIMENT task (Table 5) is adequate singleton set coverage obtained. For the in-distribution PROTEIN task, coverage falls to the $70s$ for CASP12 and the low $80s$ for TS115 for the $y = $ OTHER class (Table 4). On the low-accuracy, class-imbalanced GRAMMAROOD task (Table 5), in which the minority class occurs with a proportion less than $\alpha$, singleton set coverage for the minority class is very poor. **Re-weighting the empirical CDF near a test point is not an adequate solution to obtain singleton set coverage.** The LOCAL$_{\text{CONF}}$ approach obtains coverage on the distribution-shifted SENTIMENTOOD task (Table 5), but coverage is inadequate, as with simpler Conformal Predictors, for the GRAMMAROOD task. **The stronger per-class Mondrian Conformal guarantee is also not sufficient to obtain singleton set coverage in practice.** The ADMIT sets obtain less severe under-coverage on the GRAMMAROOD task compared to the marginal Conformal Predictors, but singleton set coverage is not clearly better on the in-distribution PROTEIN task. The under-coverage of these approaches could come as a surprise to end-users. In this way, such split-conformal approaches are not ideal for instance-level decision making.

**Fortunately, we can nonetheless achieve the desired desiderata with distribution-free methods, but we need to instead rely on Venn Predictors, and recast our goal in terms of calibration rather than coverage.** The base approach with test-point up-weighting, VENN-ADMIT, is well-calibrated across tasks. We further note that there is no cost to be paid on these datasets by rejecting points in all partitions other than that with $q = K$, as seen with VENN-ADMIT$_{qK}$. That is, the admitted points are almost exclusively in the $q = K$ partition. By means of comparison

Table 4: Selective classification evaluation on PROTEIN test sets.

| | | $|\mathcal{C}|=1$ by Class Label (Amino-Acid/Token-Level Sequence Labeling) | | | | | | | |
| | | $y = $ **H**ELIX | | $y = $ **S**TRAND | | $y = $ **O**THER | | $y \in \{$**H, S, O**$\}$ | |
| Set | Method | $\overline{y \in \mathcal{C}}$ | $\frac{n}{N}$ | $\overline{y \in \mathcal{C}}$ | $\frac{n}{N}$ | $\overline{y \in \mathcal{C}}$ | $\frac{n}{N}$ | $\overline{y \in \mathcal{C}}$ | $\frac{n}{N}$ |
|---|---|---|---|---|---|---|---|---|---|
| TS115 ($N = 29,704$) | | | | | | | | | |
| | RAPS$_{\text{SIZE}}$ | 0.96 | 0.22 | 0.88 | 0.06 | 0.82 | 0.14 | 0.90 | 0.43 |
| | RAPS$_{\text{ADAPT}}$ | 0.96 | 0.24 | 0.86 | 0.07 | 0.82 | 0.17 | 0.90 | 0.48 |
| | APS | 0.96 | 0.24 | 0.86 | 0.07 | 0.83 | 0.17 | 0.90 | 0.48 |
| | LOCAL$_{\text{CONF}}$ | 0.96 | 0.23 | 0.85 | 0.07 | 0.86 | 0.18 | 0.91 | 0.49 |
| | ADMIT | 0.96 | 0.23 | 0.88 | 0.07 | 0.76 | 0.14 | 0.88 | 0.43 |
| | VENN-ADMIT-W | 0.98 | 0.14 | 0.91 | 0.03 | 0.92 | 0.08 | 0.95 | 0.24 |
| | VENN-ADMIT$_{qK}$-W | 0.98 | 0.14 | 0.92 | 0.03 | 0.92 | 0.08 | 0.96 | 0.24 |
| | VENN-ADMIT | 0.99 | 0.14 | 0.92 | 0.03 | 0.91 | 0.07 | 0.96 | 0.24 |
| | VENN-ADMIT$_{qK}$ | 0.99 | 0.14 | 0.92 | 0.03 | 0.91 | 0.07 | 0.96 | 0.24 |
| CASP12 ($N = 7,256$) | | | | | | | | | |
| | RAPS$_{\text{SIZE}}$ | 0.96 | 0.14 | 0.85 | 0.05 | 0.77 | 0.13 | 0.87 | 0.31 |
| | RAPS$_{\text{ADAPT}}$ | 0.95 | 0.16 | 0.85 | 0.06 | 0.78 | 0.15 | 0.86 | 0.36 |
| | APS | 0.95 | 0.15 | 0.86 | 0.05 | 0.74 | 0.15 | 0.85 | 0.36 |
| | LOCAL$_{\text{CONF}}$ | 0.97 | 0.14 | 0.82 | 0.03 | 0.84 | 0.12 | 0.90 | 0.30 |
| | ADMIT | 0.94 | 0.16 | 0.87 | 0.06 | 0.67 | 0.12 | 0.83 | 0.34 |
| | VENN-ADMIT-W | 0.95 | 0.09 | 0.85 | 0.01 | 0.90 | 0.06 | 0.92 | 0.16 |
| | VENN-ADMIT$_{qK}$-W | 0.96 | 0.09 | 0.87 | 0.01 | 0.89 | 0.06 | 0.93 | 0.16 |
| | VENN-ADMIT | 0.96 | 0.09 | 0.87 | 0.01 | 0.89 | 0.06 | 0.93 | 0.16 |
| | VENN-ADMIT$_{qK}$ | 0.96 | 0.09 | 0.87 | 0.01 | 0.89 | 0.06 | 0.93 | 0.16 |

Table 5: Selective classification evaluation on distribution-shifted data. SENTIMENT (in-dist.) for contrast.

| | | $|\mathcal{C}|=1$ by Class Label (Binary **SSL** and **DC**) | | | | | |
| | | $y = 0$ | | $y = 1$ | | $y \in \{0,1\}$ | |
| Set | Method | $\overline{y \in \mathcal{C}}$ | $\frac{n}{N}$ | $\overline{y \in \mathcal{C}}$ | $\frac{n}{N}$ | $\overline{y \in \mathcal{C}}$ | $\frac{n}{N}$ |
|---|---|---|---|---|---|---|---|
| SENTIMENTOOD ($N = 4750$) | | | | | | | |
| | $f(x)_{\text{tr}}^{\text{KNN}}$ (ACC.) | 0.86 | 0.50 | 0.72 | 0.50 | 0.79 | 1.0 |
| | CONF$_{\text{BASE}}$ | 0.86 | 0.50 | 0.72 | 0.50 | 0.79 | 1.0 |
| | RAPS$_{\text{ADAPT}}$ | 0.79 | 0.27 | 0.91 | 0.33 | 0.86 | 0.61 |
| | RAPS$_{\text{SIZE}}$ | 0.75 | 0.50 | 0.80 | 0.50 | 0.78 | 1.00 |
| | APS | 0.80 | 0.28 | 0.91 | 0.33 | 0.86 | 0.61 |
| | LOCAL$_{\text{CONF}}$ | 0.90 | 0.10 | 0.96 | 0.18 | 0.94 | 0.28 |
| | ADMIT | 0.96 | 0.16 | 0.93 | 0.18 | 0.94 | 0.33 |
| | VENN-ADMIT-W | 0.96 | 0.14 | 0.94 | 0.17 | 0.94 | 0.31 |
| | VENN-ADMIT$_{qK}$-W | 0.96 | 0.14 | 0.94 | 0.17 | 0.94 | 0.31 |
| | VENN-ADMIT | 0.96 | 0.13 | 0.94 | 0.17 | 0.95 | 0.29 |
| | VENN-ADMIT$_{qK}$ | 0.96 | 0.13 | 0.94 | 0.17 | 0.95 | 0.29 |
| SENTIMENT ($N = 488$) | | | | | | | |
| | $f(x)_{\text{tr}}^{\text{KNN}}$ (ACC.) | 0.94 | 0.50 | 0.91 | 0.50 | 0.93 | 1.0 |
| | CONF$_{\text{BASE}}$ | 0.94 | 0.50 | 0.91 | 0.50 | 0.93 | 1.0 |
| | RAPS$_{\text{ADAPT}}$ | 0.97 | 0.47 | 0.96 | 0.46 | 0.96 | 0.93 |
| | RAPS$_{\text{SIZE}}$ | 0.94 | 0.50 | 0.91 | 0.50 | 0.93 | 1.00 |
| | APS | 0.96 | 0.47 | 0.95 | 0.46 | 0.96 | 0.92 |
| | LOCAL$_{\text{CONF}}$ | 0.95 | 0.43 | 0.94 | 0.44 | 0.95 | 0.88 |
| | ADMIT | 0.95 | 0.42 | 0.93 | 0.40 | 0.94 | 0.82 |
| | VENN-ADMIT-W | 0.94 | 0.38 | 0.93 | 0.40 | 0.94 | 0.78 |
| | VENN-ADMIT$_{qK}$-W | 0.94 | 0.38 | 0.93 | 0.40 | 0.94 | 0.78 |
| | VENN-ADMIT | 0.94 | 0.37 | 0.94 | 0.40 | 0.94 | 0.76 |
| | VENN-ADMIT$_{qK}$ | 0.94 | 0.37 | 0.94 | 0.40 | 0.94 | 0.76 |
| GRAMMAROOD ($N = 92597$) | | | | | | | |
| | $f(x)_{\text{tr}}^{\text{KNN}}$ (ACC.) | 0.98 | 0.93 | 0.27 | 0.07 | 0.93 | 1.0 |
| | CONF$_{\text{BASE}}$ | 0.98 | 0.92 | 0.26 | 0.06 | 0.94 | 0.99 |
| | RAPS$_{\text{ADAPT}}$ | 0.97 | 0.78 | 0.34 | 0.05 | 0.94 | 0.83 |
| | RAPS$_{\text{SIZE}}$ | 0.97 | 0.79 | 0.34 | 0.05 | 0.94 | 0.84 |
| | APS | 0.97 | 0.79 | 0.34 | 0.05 | 0.94 | 0.83 |
| | LOCAL$_{\text{CONF}}$ | 1.00 | 0.85 | 0.19 | 0.05 | 0.95 | 0.91 |
| | ADMIT | 0.93 | 0.20 | 0.77 | 0.02 | 0.92 | 0.22 |
| | VENN-ADMIT-W | 1.00 | 0.11 | 0.75 | 0.01 | 0.98 | 0.12 |
| | VENN-ADMIT$_{qK}$-W | 1.00 | 0.08 | 0.89 | $<0.01$ | 0.99 | 0.09 |
| | VENN-ADMIT | 1.00 | 0.05 | 0.92 | $<0.01$ | 0.99 | 0.05 |
| | VENN-ADMIT$_{qK}$ | 1.00 | 0.05 | 0.92 | $<0.01$ | 0.99 | 0.05 |

with the VENN-ADMIT-W and VENN-ADMIT$_{qK}$-W ablations, in which calibration is obtained by VENN-ADMIT$_{qK}$-W without weighting, we would recommend making this restriction (i.e., using VENN-ADMIT$_{qK}$) as the default approach in higher-risk settings as an additional safeguard.

# 7 CONCLUSION

The finite-sample, distribution-free guarantees of Conformal Predictors are appealing; however, the coverage guarantee is too weak for typical classification use-cases. We have instead demonstrated that the key characteristics desired for prediction sets are instead achievable by calibrating weak selective classifiers with Venn Predictors, enabled by KNN approximations of the deep networks.

REPRODUCIBILITY STATEMENT

We will provide a link to our pytorch code and replication scripts with the camera-ready version of the paper. The data and pre-trained weights of the underlying Transformers are publicly available and are further described for each experiment in the Appendix.

ETHICS STATEMENT

Uncertainty quantification is a cornerstone for trustworthy AI. We have demonstrated a principled approach for selective classification that achieves the desired desiderata in challenging settings (low accuracy, class-imbalanced, distribution shifted) under a stringent class-wise evaluation scenario. We have also shown that alternative existing distribution-free approaches do not achieve the quantities typically needed in classification settings.

Whereas the use-cases for prediction sets with marginal coverage are relatively limited, the use-cases for reliable selective classification are numerous. For example, reliable class-conditional selective classification directly applies to routing to reduce overall computation (e.g., use small, fast models, only deferring to larger models for rejected predictions), and higher-risk settings where less confident predictions must be sent to humans for further adjudication.

An unusual and advantageous aspect of a VENN-ADMIT Predictor, and which further distinguishes it from post-hoc Platt-scaling-style calibration (Platt, 1999; Guo et al., 2017), is a degree of inherent example-based interpretability: The calibrated distribution for a point is a simple transformation of the empirical probability among similar points, with partitions determined by a KNN that can be readily inspected. This matching component yields a direct avenue for addressing group-wise fairness: Known group attributes can be incorporated as categories to ensure group-wise calibration.

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

## A    APPENDIX: CONTENTS

Appendix B describes the KNN localizer, and Appendix C provides additional details for training the KNNs. Appendix D provides guidelines on controlling for—and conveying the variance of— the sample size. We provide pseudo-code in Appendix E. In Appendix F, G, and H, we provide additional details for each of the tasks.

## B    KNN LOCALIZER

We use a KNN localizer, against the calibration set, as a localized conformal (Guan, 2022) baseline of comparison, and to re-weight category assignments for the VENN-ADMIT Predictor. This KNN localizer recasts the test approximation output as a weighted linear combination over the calibration set approximations:

$$f^c(x_t)^{\text{KNN}}_{\text{tr}} \approx f^c(x_t)^{\widehat{\text{KNN}}}_{\text{ca}} = \sum_{\substack{k \in \underset{j \in \{I+1, \ldots, I+|\mathcal{D}_{\text{ca}}|\}}{\arg \text{K} \min} ||\boldsymbol{r}_t - \boldsymbol{r}_j||_2}} \psi_k \cdot f^c(x_k)^{\text{KNN}}_{\text{tr}}, \text{where } K = |\mathcal{D}_{\text{ca}}| \quad (8)$$

The single parameter, the temperature parameter of $\psi_k \in [0, 1]$, which is calculated in an analogous manner as $w_k$ in Equation 3, is trained via gradient descent against $\mathcal{D}_{\text{te}}$ to minimize *prediction* discrepancies between $f(x_t)^{\text{KNN}}_{\text{tr}}$ and $f(x_t)^{\widehat{\text{KNN}}}_{\text{ca}}$. As with $f(x)^{\text{KNN}}_{\text{tr}}$, training is performed using $\mathcal{D}_{\text{knn}}$.

As noted in the main text, we can use the weights from this approximation as a guard against distribution shifts within the data partitions. For a given test point, we calculate $f(x_t)^{\widehat{\text{KNN}}}_{\text{ca}}$ (Eq. 8), and then determine the augmented distribution (i.e., $p_c(\cdot)$, the empirical probability for a point when including the point itself, assuming a given label $c$) for the Venn Predictor by adding the test point up-weighted according to the weights of this KNN localizer. Specifically, the new weight for the test point is as follows:

$$\frac{1}{\psi'} = \frac{1}{\sum_{\{j:\, x_j \in \mathcal{T}\}} \psi_j} \,, \quad (9)$$

where $\mathcal{T}$ is the set of calibration points belonging to the same category as $x_t$. When this weight is 1, we have the standard Venn Predictor; when this weight is greater than 1, it is a sign of a mismatch (due to a distribution shift, or an otherwise aberrant category assignment) and the minimum probability estimated by the Venn Predictor becomes smaller. $\frac{1}{\psi'} \in [1, \infty)$ satisfies Prop. 3, so calibration of the selective classifier is maintained when using this weight to up-weight the test point.

## C    KNN TRAINING

We train $f(x)^{\text{KNN}}_{\text{tr}}$ and $f(x)^{\widehat{\text{KNN}}}_{\text{ca}}$ with the same learning procedure, the only difference being the underlying model that is approximated. Here, we take as $o^c$ the unnormalized output logit for class $c$ of the model to be approximated (either the MEMORY LAYER or $f(x)^{\text{KNN}}_{\text{tr}}$) and $a^c$ the unnormalized output logit for class $c$ of the approximation (either $f(x)^{\text{KNN}}_{\text{tr}}$ or $f(x)^{\widehat{\text{KNN}}}_{\text{ca}}$). The binary cross-entropy loss for a token, $t$, is then calculated as follows:

$$\mathcal{L}_t = \frac{1}{|\mathcal{Y}|} \sum_{c \in \mathcal{Y}} -\sigma(o^c) \cdot \log \sigma\left(a^c\right) - (1 - \sigma(o^c)) \cdot \log\left(1 - \sigma\left(a^c\right)\right) \quad (10)$$

That is, we seek to minimize the difference between the original model's output and the KNN's output, for each class, holding the parameters of the original model fixed. $\mathcal{L}_t$ is averaged over all classes in mini-batches constructed from the tokens of shuffled documents. We train with Adadelta (Zeiler, 2012) with a learning rate of 1.0, choosing the epoch that minimizes

$$\delta^{KNN} = \sum_{\text{DEV}} [\arg \max_{c \in \mathcal{Y}} (o) \neq \arg \max_{c \in \mathcal{Y}} (a)] \,, \quad (11)$$

the total number of prediction discrepancies between the original model and the KNN approximation over the KNN DEV set. During training, if the immediately preceding epoch did not yield a new minimal $\delta^{KNN}$ among the running epochs, we subsequently only calculate $\mathcal{L}_t$ for the tokens with prediction discrepancies until a new minimum $\delta^{KNN}$ is found (after which we return to calculating the loss over all points), or the maximum number of epochs is reached. This has a regularizing effect: There is signal in the magnitude of the KNN output, so we aim to optimize in the direction of minimizing the residuals; however, we seek to avoid over-fitting to the magnitude of the outliers.

A key insight is that we can readily approximate the vast majority of the predictions from the Transformer networks (possibly other networks, as well) using such KNN approximations, and critically, when the approximations diverge from the model, those points tend to be from the subsets over which the underlying model is itself unreliable. This implies a non-homogenous error distribution, and we find that the aforementioned procedure of iterative masking effectively learns the KNN parameters without the need to introduce other regularization approaches. In practice, we find that a relatively small amount of data (e.g., only 10% of the original validation sets for the tasks in the experiments in the main text) is sufficient to learn the low number of parameters of the KNNs.

## D  CONTROLLING FOR SAMPLE SIZE

Given a single sample from $P_{XY}$ (i.e., our single $\mathcal{D}_{ca}$ of some fixed size), we need to convey the variance due to the observed sample size. We opt for a simple hard threshold, $\kappa$, given that the distribution of split-conformal coverage is Beta distributed (Vovk, 2012). With, for example $\kappa = 1000$, assuming exchangeability, the finite-sample guarantee then implies $\approx \pm \leq 0.02$ coverage variation within a conditioning band of size $\geq 1000$ with $\alpha = 0.1, |\mathcal{D}_{te}| = \infty$. See the comprehensive tutorial Angelopoulos & Bates (2021) for additional details. In our experiments, if the size of at least 1 label-specific band for a given point falls below $\kappa$, we revert to a set of full cardinality. For the PROTEIN, SENTIMENT, and SENTIMENTOOD tasks, we set $\kappa = 1000$. With the low accuracy and low frequency of the minority class in the GRAMMAROOD task, the $q = K$ partition is comparatively small. As such, for the GRAMMAROOD task, we set $\kappa = 100$ to avoid heavily censoring the $q = K$ partition, at the expense of potentially higher variability.

## E  PSEUDO-CODE

Algorithm 1 provides pseudo-code for constructing a well-calibrated selective classification, via the two stage approach described in the text. First, Algorithm 2 constructs an ADMIT prediction set for a test point, $x_t$. If the set only includes a single class (i.e., the weak selective classifier admitted the class), the output is then calibrated via the VENN-ADMIT Predictor (Algorithm 3). The class prediction is then returned if the calibrated probability exceeds the provided threshold, $\alpha$. As an additional safeguard, we can further restrict the partitions based on $q$, as described in Section 4.4.1.

In the main text, we also compare to a variation without the test-point weighting. This unweighted variation appears in Algorithm 4 and replaces the corresponding line in Algorithm 1.

---

**Algorithm 1** VENN-ADMIT Selective Classification

---

**Input:** $\mathcal{D}_{ca}, (x_t \in \mathcal{D}_{te}, q_t, d_t)$, band radius $\omega$, $f(x)_{tr}^{KNN}, \alpha$, localizer $f(x)_{ca}^{\widehat{KNN}}$

1: **procedure** SELECTIVE-CLASSIFICATION($\mathcal{D}_{ca}, x_t, q_t, d_t, \omega, f(x)_{tr}^{KNN}, \alpha$, localizer $f(x)_{ca}^{\widehat{KNN}}$)
2:    $s \leftarrow \perp$                                          $\triangleright$ Reject option
3:    $\hat{\mathcal{C}}(x_t) \leftarrow$ ADMIT($\mathcal{D}_{ca}, x_t, q_t, d_t, \omega, f(x)_{tr}^{KNN}, \alpha$)       $\triangleright$ Mondrian Conformal prediction set (Alg. 2)
4:    **if** $|\hat{\mathcal{C}}(x_t)| = 1$ **then**
5:        $c' \leftarrow c \in \hat{\mathcal{C}}(x_t)$                              $\triangleright$ Output of the weak selective classifier
6:        $p(c') \leftarrow$ WEIGHTED-VENN-ADMIT($\mathcal{D}_{ca}, x_t, q_t, d_t, \omega, f(x)_{tr}^{KNN}, f(x)_{ca}^{\widehat{KNN}}$)
7:        **if** $p(c') \geq 1 - \alpha$ **then**
8:            $s \leftarrow c'$
**Output:** $s$, selective classification (class prediction or reject option)

---

---

**Algorithm 2** ADMIT Prediction Sets via Neural Model Approximations

---

**Input:** $\mathcal{D}_{\mathrm{ca}}, (x_t \in \mathcal{D}_{\mathrm{te}}, q_t, d_t)$, band radius $\omega$, $f(x)_{\mathrm{tr}}^{\mathrm{KNN}}, \alpha$
1: **procedure** _THRESHOLD$(\mathcal{I}', \alpha)$          $\triangleright$ Standard split-conformal if $\mathcal{I}' = \mathcal{D}_{\mathrm{ca}}$
2:       $\mathcal{S}_j \leftarrow s(x_j) = 1 - \hat{\pi}^y(x_j)_{\mathrm{tr}}^{\mathrm{KNN}}, \forall\, x_j \in \mathcal{I}'$      $\triangleright$ Conformity scores over calibration subset
3:       $\hat{l}^\alpha \leftarrow \lceil (|\mathcal{I}'| + 1)(1 - \alpha) \rceil / |\mathcal{I}'|$ quantile of $\mathcal{S}$
4:       **return** $\hat{\tau}^\alpha \leftarrow 1 - \hat{l}^\alpha$
5: **procedure** ADMIT$(\mathcal{D}_{\mathrm{ca}}, x_t, q_t, d_t, \omega, f(x)_{\mathrm{tr}}^{\mathrm{KNN}}, \alpha)$
6:       $\hat{\mathcal{C}}(x_t) \leftarrow \{\hat{y}_t^{\mathrm{KNN}}\}$
7:       $\mathcal{I} \leftarrow \mathcal{B}(x_t, \omega, q_t, d_t; \mathcal{D}_{\mathrm{ca}})$        $\triangleright$ Calibration points in band centered at $x_t$ (Eq. 6)
8:       **for** $c \in \{1, \ldots, C\}$ **do**
9:           $\mathcal{I}^c \leftarrow \{x_j : x_j \in \mathcal{I}, y_j = c\}$        $\triangleright$ Subset of band for which true class is $c$
10:          $\hat{\tau}_c^\alpha \leftarrow$ _THRESHOLD$(\mathcal{I}^c, \alpha)$
11:          $\hat{\mathcal{C}}(x_t) \leftarrow \hat{\mathcal{C}}(x_t) \cup \big\{ c : \hat{\pi}^c(x_t)_{\mathrm{tr}}^{\mathrm{KNN}} \geq \hat{\tau}_c^\alpha \big\}$
12:       **return** $\hat{\mathcal{C}}(x_t)$
**Output:** $\hat{\mathcal{C}}(x_t)$, prediction set

---

**Algorithm 3** Conservative Calibration via Inductive VENN-ADMIT Predictor (weighted)

---

**Input:** $\mathcal{D}_{\mathrm{ca}}, (x_t \in \mathcal{D}_{\mathrm{te}}, q_t, d_t)$, band radius $\omega$, $f(x)_{\mathrm{tr}}^{\mathrm{KNN}}$, localizer $f(x)_{\mathrm{ca}}^{\widehat{\mathrm{KNN}}}$
1: **procedure** _CATEGORY$(\mathcal{I}, x_t, y')$          $\triangleright$ Same as in Alg. 4
2:       $\mathcal{T} \leftarrow \{(x_t, y')\}$         $\triangleright$ $y'$ is the *assumed* true label for $x_t$
3:       **for** $x_j \in \mathcal{I}$ **do**
4:          **if** $\hat{y}_t^{\mathrm{KNN}} = \hat{y}_j^{\mathrm{KNN}} \wedge \hat{\mathcal{C}}(x_t) = \hat{\mathcal{C}}(x_j)$ **then**      $\triangleright$ $\hat{\mathcal{C}}$ calculated as in Alg. 2
5:            $\mathcal{T} \leftarrow \mathcal{T} \cup \{(x_j, y_j)\}$        $\triangleright$ $y_j$ is the true label for $x_j$
6:       **return** $\mathcal{T}$
7: **procedure** WEIGHTED-VENN-ADMIT$(\mathcal{D}_{\mathrm{ca}}, x_t, q_t, d_t, \omega, f(x)_{\mathrm{tr}}^{\mathrm{KNN}}, f(x)_{\mathrm{ca}}^{\widehat{\mathrm{KNN}}})$
8:       $\mathcal{I} \leftarrow \mathcal{B}(x_t, \omega, q_t, d_t; \mathcal{D}_{\mathrm{ca}})$        $\triangleright$ Calibration points in band centered at $x_t$ (Eq. 6)
9:       **for** $c \in \{1, \ldots, C\}$ **do**
10:       $\mathcal{T} \leftarrow$ _CATEGORY$(\mathcal{I}, x_t, c) \setminus \{(x_t, c)\}$       $\triangleright$ Exclude $x_t$ from the category
11:       $\psi' \leftarrow \sum_{\{j : x_j \in \mathcal{T}\}} \psi_j$       $\triangleright$ Sum of weights from KNN localizer Eq. 8; $\psi' \in (0, 1]$
12:          **for** $c' \in \{1, \ldots, C\}$ **do**
13:            $p_c(c') \leftarrow \dfrac{|\{(x^*, y^*) \in \mathcal{T} : y^* = c'\}| + [c = c'] \cdot (\frac{1}{\psi'})}{|\mathcal{T}| + \frac{1}{\psi'}}$      $\triangleright$ Test-point weighted empirical probability
14:       **for** $c' \in \{1, \ldots, C\}$ **do**
15:          $\underline{p}(c') \leftarrow \min_{c \in C} p_c(c')$      $\triangleright$ Lower Venn probability for each class (across augmented distributions)
16:       **return** $\underline{p}(\cdot)$
**Output:** $\underline{p}(\cdot)$, lower Venn calibrated distribution for $x_t$.

---

# F    TASK: PROTEIN SECONDARY STRUCTURE PREDICTION (PROTEIN)

In the supervised sequence labeling PROTEIN task, we seek to predict the secondary structure of proteins. For each amino acid, we seek to predict one of three classes, $y \in$ {HELIX, STRAND, OTHER}.

For training and evaluation, we use the TAPE datasets of Rao et al. (2019).[3] We approximate the Transformer of Rao et al. (2019), which is *not* SOTA on the task; while not degenerate, this fine-tuned self-supervised model was outperformed by models with HMM alignment-based input features in the original work. Of interest in the present work is whether coverage can be obtained with a neural model with otherwise relatively modest overall point accuracy. We use the publicly available model and pre-trained weights[4].

---

[3]TAPE provides a standardized benchmark from existing models and data (El-Gebali et al., 2019; Berman et al., 2000; Moult et al., 2018; Klausen et al., 2019).

[4]https://github.com/songlab-cal/tape

---

**Algorithm 4** Conservative Calibration via Inductive VENN-ADMIT Predictor (unweighted)

---

**Input:** $\mathcal{D}_{\mathrm{ca}}, (x_t \in \mathcal{D}_{\mathrm{te}}, q_t, d_t)$, band radius $\omega$, $f(x)_{\mathrm{tr}}^{\mathrm{KNN}}$

1: **procedure** _CATEGORY$(\mathcal{I}, x_t, y')$
2:      $\mathcal{T} \leftarrow \{(x_t, y')\}$                                $\triangleright$ $y'$ is the *assumed* true label for $x_t$
3:      **for** $x_j \in \mathcal{I}$ **do**
4:          **if** $\hat{y}_t^{\mathrm{KNN}} = \hat{y}_j^{\mathrm{KNN}} \wedge \hat{\mathcal{C}}(x_t) = \hat{\mathcal{C}}(x_j)$ **then**          $\triangleright$ $\hat{\mathcal{C}}$ calculated as in Alg. 2
5:              $\mathcal{T} \leftarrow \mathcal{T} \cup \{(x_j, y_j)\}$                  $\triangleright$ $y_j$ is the true label for $x_j$
6:      **return** $\mathcal{T}$
7: **procedure** VENN-ADMIT$(\mathcal{D}_{\mathrm{ca}}, x_t, q_t, d_t, \omega, f(x)_{\mathrm{tr}}^{\mathrm{KNN}})$
8:      $\mathcal{I} \leftarrow \mathcal{B}(x_t, \omega, q_t, d_t; \mathcal{D}_{\mathrm{ca}})$          $\triangleright$ Calibration points in band centered at $x_t$ (Eq. 6)
9:      **for** $c \in \{1, \dots, C\}$ **do**
10:        $\mathcal{T} \leftarrow$ _CATEGORY$(\mathcal{I}, x_t, c)$
11:        **for** $c' \in \{1, \dots, C\}$ **do**
12:            $p_c(c') \leftarrow \frac{|\{(x^*, y^*) \in \mathcal{T} : y^* = c'\}|}{|\mathcal{T}|}$            $\triangleright$ Empirical probability
13:      **for** $c' \in \{1, \dots, C\}$ **do**
14:        $\underline{p}(c') \leftarrow \min_{c \in C} p_c(c')$     $\triangleright$ Lower Venn probability for each class (across augmented distributions)
15:      **return** $\underline{p}(\cdot)$

**Output:** $\underline{p}(\cdot)$, lower Venn calibrated distribution for $x_t$.

---

### F.1 MEMORY LAYER

The base network consists of a pre-trained Transformer similar to BERT$_{\mathrm{BASE}}$ with a final convolutional classification layer, consisting of two 1-dimensional CNNs: The first over the final hidden layer of the Transformer corresponding to each amino acid (each hidden layer is of size 768), using 512 filters of width 5, followed by ReLU and a second CNN using 3 filters of width 3. Batch normalization is applied before the first CNN, and weight normalization is applied to the output of each of the CNNs. The application of the 3 filters of the final CNN produces the logits, $\mathbb{R}^3$, for each amino acid.

The MEMORY LAYER consists of an additional 1-dimensional CNN, which uses 1000 filters of width 1. The input to the MEMORY LAYER corresponding to each amino acid is the concatenation of the final hidden layer of the Transformer, the output of the final CNN of the base network, and a randomly initialized 10-dimensional word-embedding. The output of the CNN is passed to a LinearLayer of dimension 1000 by 3. (Unlike the sparse supervised sequence labeling task of GRAMMAROOD, we use neither a ReLU, nor a max-pool operation. The sequences are very long in this setting—up to 1000 used in training and 1632 at inference to avoid truncation—so removing the max-pool bottleneck enables keeping the number of filters of the CNN lower than the total number of amino acids. In this way, we also do not use the decomposition of the CNN with the LinearLayer, as in the GRAMMAROOD task, since the sparsity over the input is not needed for this task.) The exemplar vectors for the KNNs are then the $r \in \mathbb{R}^{1000}$ filter applications of the CNN corresponding to each amino acid.

We fine-tune the base network and train the MEMORY LAYER in an iterative fashion. Each epoch we either update the gradients of the base network, or those of the MEMORY LAYER, freezing the counter-part each epoch. We start by updating the base network (and freezing the MEMORY LAYER), and we use separate optimizers for each: Adadelta (Zeiler, 2012) with a learning rate of 1.0 for the MEMORY LAYER and Adam with weight decay (Loshchilov & Hutter, 2019) with a learning rate of 0.0001 and a warmup proportion of 0.01 for the base network. For the latter, we use the BertAdam code from the HuggingFace re-implementation of Devlin et al. (2019). We fine-tune for up to 16 epochs, and we use a standard cross-entropy loss.

## G SUPERVISED GRAMMATICAL ERROR DETECTION (GRAMMAROOD)

The GRAMMAROOD task is a binary sequence labeling task in which we aim to predict whether each word in the input does ($y_t = 1$) or does not ($y_t = 0$) have a grammatical error. $\mathcal{D}_{\mathrm{tr}}$ and $\mathcal{D}_{\mathrm{ca}}$ consist of essays written by second-language learners (Yannakoudakis et al., 2011; Rei & Yan-

nakoudakis, 2016) and $\mathcal{D}_{\text{te}}$ consists of student written essays *and* newswire text (Chelba et al., 2014). The test set is the FCE+NEWS2K set of Schmaltz (2021).

The test set is challenging for two reasons. First, the $y = 1$ class appears with a proportion of 0.07 of all of the words. This is less than our default value for $\alpha$, with the implication that marginal coverage can potentially be obtained by altogether ignoring that class. Second, the in-domain task itself is relatively challenging, but it is made yet harder by adding newswire text, as evident in the large $F_{0.5}$ score differences across $\mathcal{D}_{\text{ca}}$ and $\mathcal{D}_{\text{te}}$ in Table 2.

The exemplar vectors, $r \in \mathbb{R}^{1000}$, used in the KNNs are extracted from the filter applications of a penultimate CNN layer over a frozen $\text{BERT}_{\text{LARGE}}$ model, as in Schmaltz (2021).

## H   TASKS: SENTIMENT CLASSIFICATION (SENTIMENT) AND OUT-OF-DOMAIN SENTIMENT CLASSIFICATION (SENTIMENTOOD)

SENTIMENT and SENTIMENTOOD are document-level binary classification tasks in which we aim to predict whether the document is of negative ($y = 0$) or positive ($y = 1$) sentiment. The training and calibration sets, as well as the base networks, are the same for both tasks, with the distinction in the differing test sets. The training set is the 3.4k IMDb movie review set used in Kaushik et al. (2020) from the data of Maas et al. (2011). For calibration, we use a disjoint 16k set of reviews from the original training set of Maas et al. (2011). The test set of SENTIMENT is the 488-review in-domain test set of original reviews used in Kaushik et al. (2020), and the test set of SENTIMENTOOD consists of 5k Twitter messages from SemEval-2017 Task 4a (Rosenthal et al., 2017).

Similar to the GRAMMAROOD task, the exemplar vectors, $r \in \mathbb{R}^{2000}$, are derived from the filter applications of a penultimate CNN layer over a frozen $\text{BERT}_{\text{LARGE}}$ model. However, in this case, the vectors are the concatenation of the document-level max-pooled vector, $r \in \mathbb{R}^{1000}$, and the vector associated with a single representative token in the document, $r \in \mathbb{R}^{1000}$. To achieve this, we model the task as multi-label classification and fine-tune the penultimate layer CNN and a final layer consisting of two linear layers with the combined min-max and global normalization loss of Schmaltz & Beam (2020). In this way, we can associate each word with one of (or in principle, both) positive and negative sentiment, or a neutral class, while nonetheless having a single exclusive global prediction. This provides sparsity over the detected features, and captures the notion that a document may, in totality, represent one of the classes (e.g., indicate a positively rated movie overall) while at the same time including sentences or phrases that are of the opposite class (e.g., aspects that the reviewer rated negatively). This behavior is illustrated with examples from the calibration set in Table 6. We use the max scoring word from the "convolutional decomposition", a hard-attention-style approach, for the document-level predicted class as the single representative word for the document. For the document-level prediction, we take the max over the multi-label logits, which combine the global and max local scores.

Table 6: Model feature detections from snippets from $\mathcal{D}_{\text{ca}}$ for SENTIMENT and SENTIMENTOOD, for which prediction sets are constructed for the binary document-level predictions. Most documents only have features of a single class detected (as in the example in the final row), but our modeling choice (Section H) does enable multi-label detection as in the first example, for which the true document label is `positive sentiment`, and the second example, for which the true document label is `negative sentiment`. The max scoring word for each document is underlined.

| Model predictions over $\mathcal{D}_{\text{ca}}$ |
|---|
| What `an` `amazing` film. *[...]* My only gripe is that it has not been released on video in Australia and is therefore only available on TV. What a `waste.` |
| *[...]* But the story that then develops `lacks` any of the stuff that these opening fables display. *[...]* I will say that the music by Aimee Mann was `great` and I'll be looking for the Soundtrack CD. *[...]* |
| Kenneth Branagh `shows` off his `excellent` skill in both acting and writing in this `deep` and thought provoking interpretation of Shakespeare's most classic and well-written tragedy. *[...]* |

