# OpenReview forum: "Approximate Conditional Coverage via Neural Model Approximations"
_ICLR.cc/2023/Conference — Submitted to ICLR 2023_

### Official Review · Reviewer_ZHj3 · 2022-10-24

**Confidence:** 4
**Clarity, Quality, Novelty And Reproducibility:** See above.
**Correctness:** 3
**Technical Novelty And Significance:** 2
**Empirical Novelty And Significance:** 2
**Recommendation:** 3

**Strength And Weaknesses:**

Strengths:
- The paper addresses an important problem of improving approximate conditional coverage.
- It is generally ambitious to demand conditional coverage not only in neighborhoods but also depending on label _and_ the prediction set size. The later, in particular, is known to be difficult and sometimes hinders interpretation of uncertainty for individual examples.
- The introduction clearly states this motivation and I also like the short overview of the NLP tasks tackled.
- The paper includes experiments on several interesting datasets and Table 1 helps to get an overview of the tasks.

Weaknesses:
Writing:
In general, I feel that the method is very difficult to follow. This is mainly due to a missing high-level overview of how individual methods are put together, what is trained/calibrated on which set and how it comes together. It is made worse by an extremely confusing description of VENN predictors that are used on top of standard conformal prediction sets.
Individual comments:
- The paper seems to highlight to work on transformers, which is fine. However, I find that the method itself should be independent of the architecture, so I find this emphasize confusing at times. In my opinion, writing would benefit from assuming a general function that produces features for tokens or documents depending on the task. Or am I overlooking anything transformer-specific?
- I had difficulties following the KNN approximation introduced in 4.1. This is partly due to notation (is arg K min the K-minimum elements over I?) and some ambiguity how the parameters are actually learned. Moreover, the definition of q_t – the “count of consecutive sign matches” is not very intuitive. Is the argument really capital K, or should it be lower-case k? Why is q_t assumed to be useful?
- Based on the appendix, in 4.2, standard conformal calibration is used to obtain thresholds. This is not really clear from 4.2 and I feel the main paper would benefit from the algorithms and a discussion.
- There is no real introduction of VENN predictors which really makes 4.3 difficult if not impossible to follow without reading up on it. Footnote 1 is redundant and not helpful.
- Footnote 2 is also a bit vague. So leave-one-out-cross-validation is used to obtain a prediction set for each calibration set. But what implication does this have in terms of a coverage guarantee? Especially as it is stated to be explicitly different from cross-conformal or Jacknife+ conformal prediction.
- The third paragraph in 4.3 is entirely unclear to me – what are these augmented distributions and points?
- Footnote 4 is also a bit unmeaning.
- Tables 3, 4 and 5 are very packed and in my opinion poorly introduced/described. Throughout the text it remains unclear what exactly is shown and how I should read these tables in order to draw the same conclusions as the authors.

Method:
- The method requires an additional training set for the KNN, this is not really ablated or discussed. Compared to other methods working without the KNN, does that introduce a disadvantage?
- What is the intuition of the additional mass calibration?
- Regarding contributions, I am a bit unsure whether the main contribution is the combination of the KNN approach with conformal prediction and VENN predictors or whether there is any methodological contribution in the VENN predictors itself. Could the authors clarify? For me it looks like the main contribution is the combination of these approaches, but this combination is partly difficult to follow due to poor writing.
- What kind of guarantee do I get from this approach. Due to confusion of the leave-one-out conformal predictor, I am unsure whether a standard marginal or even class-conditional guarantee is obtained in addition to better empirical conditional coverage.

Experiments:
- In general, I do not understand how experiments are run. Are the reported numbers averaged across different KNN/cal/test splits on these datasets? If so, how many samples have been used and why not plot or report the variation in coverage and size?
- Why is RAPS applied on the transformer but VENN-ADMIT on the KNN? I am missing a KNN-based RAPS baseline and a discussion why VENN-ADMIT does not work directly on the KNN.
- In Table 3, I cannot really see where the “empirical behavior of the calibration points differs significantly”. Across most parts, numbers seem to be roughly similar across some labels. What am I supposed to compare?

**Summary Of The Paper:**

The authors propose a KNN-based Venn-predictor for improved approximate coverage for text and token classification tasks.

**Summary Of The Review:**

I do not believe this paper to be ready for publications. This is mainly based on confusing writing, that makes it difficult to understand not only the method but also the provided guarantees and judge novelty and experiments.

---

> ### Author Response · Authors · 2022-11-19
> **Additional notes**
>
> We thank the reviewers for their comments and suggestions. We have revised the manuscript, as noted in the message above. We briefly include some additional points here.
>
> --
>
> It seems plausible that the overall behavior will be true of other deep networks, but we focus our empirical experiments in the current work on Transformers given their ubiquity in NLP tasks, including in real-world settings.
>
> We have added additional details on training the KNNs in Appendix C. We provide additional intuition for the q feature (which is a key component of the approach) in the response to Reviewer dUK3 above.
>
> There is a cost to be paid in statistical efficiency, at least in principle, since we use additional disjoint splits. (In the updated manuscript, as noted above, we tighten the theoretical argument by just using disjoint splits for the Mondrian Conformal Predictor and the Venn Predictor, with the former being treated as underlying weak selective classification models. We defer the leave-one-out approach, which incidentally does not affect the empirical results, to future work.) In practice, we find that our quantity of interest is still obtained at the given sample sizes, since the vast majority of the more reliable points still have large data partitions. The expected variance is Discussed in Appendix D. In the experiments, we rely on multiple data sets and tasks, with generally large evaluation sets, to assess whether our quantities are obtained.
>
> "KNN-based RAPS baseline": The coverage is no better (as in closer to conditional) when using the KNN output softmax than when using the memory layer softmax with the APS-style scores, so we omit it for space reasons. That is, the primary benefit in using the KNNs for uncertainty (and paying the additional computational overhead) comes from making use of these exogenous signals q and d.
>
> "Venn-ADMIT does not work directly on the KNN": One could construct a taxonomy just using the predicted class of the KNN or base model output. The resulting Predictor would be well-calibrated, but the quantity would not be sufficiently sharp for typical applications. This is likely a contributing reason why Venn Predictors (which are from the early 2000s) have been largely overlooked as a calibration method: It is typically not so obvious how to partition the data with standard scoring classifiers in a manner that balances specificity and generality in a satisfactory way.
>
> This latter aspect gets to the core of the novelty of the approach: The Venn-ADMIT approach defines a taxonomy that partitions the high-dimensional input of NLP tasks without having explicit attributes known in advance. Because the signals used for defining this taxonomy, q and d, encode strong signals for prediction reliability including over certain types of distribution shifts, we then also gain a degree of robustness. It is sufficiently sharp to produce informative output even over low-accuracy, class-imbalanced settings. We are not aware of an alternative calibration approaches that would have these characteristics.

---

### Official Review · Reviewer_AM2c · 2022-10-25

**Confidence:** 3
**Correctness:** 3
**Technical Novelty And Significance:** 3
**Empirical Novelty And Significance:** 3
**Recommendation:** 5

**Clarity, Quality, Novelty And Reproducibility:**

The writing of this work is clear. However, the quality could be improved if the above mentioned issues could be properly address. The idea is novel. The reproducibility is not clear as the code is not provided, plus there is no paragraph of reproducibility statement as suggested by ICLR.

**Strength And Weaknesses:**

1. I am confused why the proposed method makes sense. Usually, such new algorithms should be justified through proper theoretical guarantee from a statistical perspective. For example, predictive inference-based methods should at least provide results on valid (unconditional) coverage and conditional coverage.

2. Why is the proposed method still valid under distribution shift? Previous works need to assume the knowledge of the exact form of shifting to guarantee the exchangeability, while in this work there is no such assumption.

3. Is it possible that the proposed is trading performance with computation? A time-complexity analysis is missed.

**Summary Of The Paper:**

This work introduces a new conformal prediction-based approach for constructing calibrated predictive intervals. The improvements are from the reliability of KNN-based approximation and a novel Venn predictor. The proposed approach is evaluated on large scale sequence modeling tasks.

**Summary Of The Review:**

The proposed method in this work is novel and shows promising empirical results. I would consider raising the score if my concerns could be addressed properly.

---

> ### Author Response · Authors · 2022-11-19
> **Additional justification**
>
> We thank the reviewers for their comments and suggestions. We have revised the manuscript, as noted in the message above. We briefly include some additional points here.
>
> --
>
> 1. We have tightened the theoretical argument in the text, with respect to the main approach (relying on the lower probability of the Venn Predictor being conservatively well-calibrated), and justification for the additional heuristics.
>
> 2. This aspect of the approach is rather interesting: We can obtain at least some degree of robustness to shifts without re-weighting via the likelihood ratio/etc. To do so, the approach does indeed still make assumptions w.r.t. particular shifts: The Mondrian-exchangeability model wherein points are required to be exchangeable within partitions of the data. In this way, the approach can inherently handle changes in the proportions of the data partitions since the matching of the Venn Predictor (via categories) is done independently across data partitions. To the degree that corresponds to the shifts seen in NLP tasks (where distribution/domain shifts are in general hard to characterize) will depend on the task and data, but the empirical evidence we show (on SentimentOOD and GrammarOOD) suggests the answer is yes, in at least some cases. This reflects that the features q and d encode strong signals for prediction reliability, as seen in Table 3.
>
> 3. Yes, in a sense. APS-style conformal scores in themselves are computationally trivial to calculate, whereas we require the additional space/time of training of an additional CNN (memory layer), training of KNNs, storing dense vectors, and the concomitant dense matching at inference. However, we note that this process is still achievable in the batch setting even with exact search using a single more recent GPU. We say, "in a sense", because with our approach, we are able to reliably obtain a rather more useful quantity (with guarantees) in the classification setting, whereas the alternative (computationally cheaper) approaches do not achieve this quantity, nor exhibit the robustness over distribution shifts.
>
> As noted above, we have added the ICLR Reproducibility statement and will be releasing our code and replication scripts, as is standard.

---

### Official Review · Reviewer_sxyD · 2022-10-25

**Confidence:** 3
**Correctness:** 3
**Technical Novelty And Significance:** 2
**Empirical Novelty And Significance:** 2
**Recommendation:** 3

**Clarity, Quality, Novelty And Reproducibility:**

(Clarity) As mentioned in Weaknesses, the method section is unclear.

(Quality) The proposed approach does not come with a rigorous guarantee on conditional coverage.

(Novelty) The proposed approach is the combination of known approaches; but the combination seems novel. However, empirically, the proposed approach looks as good as baselines (as mentioned in Weaknesses), so I’m not sure if the novelty is meaningful.


**Strength And Weaknesses:**

**Strengths**:
* Extensive evaluation on NLP tasks

**Weaknesses**:
* A proposed approach is unclear (e.g., due to not well-defined terms)
* A proof on coverage guarantee is missing
* An interpretation on empirical coverage results is unusual.



(Weakness 1)
The Method section is unclearly written. First, “ADMIT” prediction sets and “Venn-ADMIT Predictor” are not well defined — probably either adding simplified algorithm blocks in the main paper (instead of Appendix) or defining the prediction sets for each type may help. Moreover, the description on Venn Predictor is missing, but pointing to an external reference; I think the paper needs to be self-contained by providing a brief summary.

(Weakness 2)
I think the main claim of this paper is that the proposed approach achieves a conditional coverage guarantee. However, its proof is missing. Considering that the conformal prediction work mainly focuses on proving correctness guarantees, this work needs rigorous proofs on the conditional guarantee. Otherwise, this work only provides empirical evidence, which is interesting, but not enough.

(Weakness 3)
Table 4 and 5 show the empirical coverage of baselines and also the proposed approach. Given alpha=0.1, the empirical coverage of conformal prediction is around 0.1 (mainly because the empirical coverage of conformal prediction sets is usually evaluated, assuming that it is conditioned on a calibration for practical reasons); thus, baseline coverage results look okay with me. I cannot see why 0.88 or 0.89 empirical coverage is bad for baselines.


**Summary Of The Paper:**

The paper proposes a new approach for prediction set construction for transformer networks, which is mainly designed to obtain approximate conditional coverage. The proposed approach is evaluated in NLP classification tasks to show its efficacy.

**Summary Of The Review:**

I think that the method section writing needs to be improved, the conditional coverage guarantee of the proposed approach needs to be proved, and the proposed approach requires to outperform the baseline (in terms of conditional coverage) to show its efficacy. Thus, currently, I vote for rejection.

---

> ### Author Response · Authors · 2022-11-19
> **Additional context w.r.t. marginal coverage**
>
> We thank the reviewers for their comments and suggestions. We have revised the manuscript, as noted in the message above. We briefly include some additional points here.
>
> --
>
> If one seeks marginal coverage, indeed 0.88 or 0.89, is within the expected variation (due to the Beta distribution of conformal coverage). However, it is the rare case that one would seek as their end goal marginal coverage for classification with the high-accuracy deep neural network models. An impossibility results precludes true conditional coverage in the distribution-free setting with finite sample, so more recent works seek other means of obtaining some notion of coverage between marginal and conditional coverage. We show that these adaptive approaches tend to still be relatively weak in practice, and we provide an alternative approach that more directly targets the quantity typically needed for classification.
>
> To put it another way, if we want to remain in the distribution-free setting, some type of compromise has to be made since conditional coverage is not achievable. The compromise we make is that we're not going to make any guarantees on the proportion of points that we admit, but of those non-rejected predictions, we guarantee (in the calibration validity sense) that they will be accurate with a frequency of at least 1-alpha.

---

### Official Review · Reviewer_dUK3 · 2022-10-27

**Confidence:** 3
**Correctness:** 3
**Technical Novelty And Significance:** 3
**Empirical Novelty And Significance:** 2
**Recommendation:** 3

**Clarity, Quality, Novelty And Reproducibility:**

Overall I think the paper is poorly written and hard to understand the main message. I have laid out the problems I encountered in the review above and hope that the authors could add more intuition into their paper.

**Strength And Weaknesses:**

Strengths:
- The paper looks at a new way to use CP in the transformer setting.
- But firstly, they can construct a novel method by approximating the transformer using a KNN and then subsequently using this approximation to perform CP.


Weakness:
- Unfortunately, the paper is very poorly written. I barely understand the method even as a person that has published in CP before.
- For example, I don't understand the data partitioning 4.1.2. there is no intuition on what is happening, and the notation seems non-understandable. q_t is defined as a feature but then it is also a function(q_t(k-1)) ? Could the authors please add a simple example for the reader to understand the intuition behind it?
- There is no explanation on Venn Arber in the paper. 4.3 is not self contained and it is hard to understand what is happening without having knowledge on this from the get go. A paper should be self contained or at least should intuition on what is happening in the algorithm.
- Also the notion of memory layer is not well explained in the paper at all hence assuming this knowledge from the reader.
- Following all of the above I have big difficulties reading this paper and getting any intuition on what the paper is trying to achieve.
- The experiments are also explained barebones and I don't understand the main message behind it. what does "Only the VENN-ADMIT sets consistently obtain acceptable singleton set coverage across datasets".

**Summary Of The Paper:**

This paper aims to construct approximate conditional coverage models using KNN (which approximates a transformer model). By firstly partitioning the data and then applying CP conditioned on the label and the partition they derive an algorithm which allows them to obtain approximate conditional coverage guarantees.

**Summary Of The Review:**

I believe the paper has to be rewritten significantly in order to be accepted. As someone who has published in the field of CP before it is even hard for me to understand what is happening in this paper. Hence I recommend the authors to add additional intuition into their paper to make their goal as well as methodology clearer.

If they authors are able to update the script with clarity I would be willing to raise my score.

---

> ### Author Response · Authors · 2022-11-19
> **Additional high-level intuition for the distribution-free context**
>
> We thank the reviewers for their comments and suggestions. We have revised the manuscript, as noted in the message above.
>
> --
>
> We have re-cast the presentation in terms of the minimal quantity one would typically seek in quantifying uncertainty in the classification setting with deep networks: well-calibrated selective classification, or similarly in the hypothesis testing framework of Conformal Predictors, singleton-set coverage. We have clarified the distinction between Conformal Predictors, Mondrian Conformal Predictors, and the lesser known (and distinct) concept of Venn Predictors. In the initial version we viewed the Venn Predictor calibration as a means of screening under-coverage of the Mondrian Conformal Predictor: Redistributing the probability mass as we did can only increase the set size, so the conservative coverage (lower bound) is not affected. In the current version, we instead view the Mondrian Conformal Predictor as a weak selective classifier (i.e., an underlying ML model), which is then calibrated via the Venn Predictor. The takeaway is similar for some applications, but the advantage is that we can interpret the output in terms of probabilities (rather than coverage), and in principle, the un-coarsened probability could be used, as well, which may be useful for some applications that need additional resolution in the output.
>
> Figure 1 provides some additional context of how the components fit together.
>
> The feature q turns out to be central. Figure 1.b. provides visual intuition: The feature q can be viewed as an ensemble between the test instance and the predictions and true labels from similar instances from training. Greater agreement suggests greater confidence in the prediction. We use the (overloaded) notation q(K) to indicate that it is parameterized by K (upper-case), the depth of the KNN using the training set as the support set. That is, the maximum value is q=K. We also use this notation to emphasize that it is calculated based on consecutive nearest matches (right side of Eq. 4 in the updated manuscript). This latter aspect is important; points with the same value of q (with increasing depth into the training/support set) tend to behave similarly in terms of point estimates. Table 3 shows how point accuracy, for example, differs markedly with points with q=0 vs. q=K. In fact, this observation is likely to be independently quite interesting in itself to the ICLR community. There is a sense in which we get a "built-in" ensemble with the deep networks that we can back-out via dense matching. In practice, we find that calibration is relatively stable in the q=K partition even over the distribution shifts observed in these datasets, whereas the calibration itself tends to be more noisy in the q<K partitions over distribution shifts. We can then further divide the space with the distance to training band; points that are similarly distant from training also tend to behave similarly to each other in terms of point estimates.
>
> Venn Predictor calibration is a matching procedure; we gain inherent robustness to changes in the proportion of our data partitions since calibration is performed independently across partitions of similar points. The next obvious question is then, "What about changes in the distribution within these data partitions?" We take a defensive approach with the additional KNN localizer (using 1/psi'): We adjust the probability for dissimilar category assignments. Due to the simple but useful Proposition 3, our notion of conservative calibration is maintained.
>
> In this way, we divide the data in terms of q, d, and (optionally) account for a notion of similarity to calibration (1/psi'). These are all calculated via the dense representations. Can we also make a division of the output space to get an even sharper calibration? Calibration approaches such as isotonic regression or histogram binning amount to dividing the output space. That's where the Mondrian Conformal Predictors come in. They end up being a useful approach for dividing the output: They simply subset based on the quantile cutoffs across labels. We can then reliably calibrate the singleton-sets/non-rejected predictions that have survived this winnowing, because points with similar q, d, 1/psi', and the same upper quantiles across labels tend to behave similarly.

---

> > ### Comment · Reviewer_dUK3 · 2022-12-11
> > **Response**
> >
> > First of all I would like to apologise for the late reply. I was away for a personal matter and then fell sick until just now ...
> >
> > I would like to give the authors some constructive feedback in terms of writing rebuttals, especially when addressing bullet points of reviewer concerns.
> >
> > If I read correctly, you have not in fact replied to my bullet points e.g. memory layer question (4th bullet point) or experiments (6th bullet point) and in terms of clarity, I am still confused with the above even after reading it multiple times.
> >
> > I do believe the paper has strengths however the clarity of the paper hinders me to see the true potential of this paper.
> >
> > Therefore I cannot increase my score.
> >
> > Best

---

> > > ### Author Response · Authors · 2022-12-12
> > > **Additional clarifications/intuition (Part I of II)**
> > >
> > > (Part I of II) We thank the reviewer for their comments. To re-state:
> > >
> > > ### memory layer question (4th bullet point):
> > >
> > > As noted on p. 3, the Memory Layer is a kernel width 1 convolutional neural network over the output representations. The exemplar vectors are then the filter applications corresponding to each input word or amino-acid. These vectors serve as distilled representations of the large network.
> > >
> > > The Appendix describes this in more detail, given space limitations of the main text. Appendix F.1 describes the Memory Layer for the Protein task. For GrammarOOD, the Memory Layer is constructed in the same manner as in the cited work: The inputs to the CNN are the top 4 hidden layers of BERT_large concatenated with the pre-trained Word2Vec word embeddings of Mikolov et al. (2013). For Sentiment and SentimentOOD the input to the CNN is the same as for GrammarOOD. The difference between GrammarOOD and the sentiment tasks is that the latter are document-level classification tasks, whereas GrammarOOD is a word-level classification task (a.k.a., sequence labeling). To get the document-level exemplar vector for the sentiment tasks, we concatenate the max-pooled vector of the CNN's filter applications (over all words) with the filter applications corresponding to 1 representative word detected by the attention mechanism (as noted in Appendix H, and illustrated in Table 6 by the underlined words). In this way, we can combine both the global and local representations for document-level classification.
> > >
> > > In practice, the Memory Layers are straightforward to train, since the parameters of the input deep network stay frozen, and the KNN's are straightforward to train given the small number of parameters. Before deploying on a new task, we can further verify the representations will be useful to this end by using the held-out validation set to check the KNN approximation matches the effectiveness of the original model (see, e.g., Table 2) and the data partitions exhibit the expected behavior as shown in Table 3.

---

> > > > ### Author Response · Authors · 2022-12-12
> > > > **Additional clarifications/intuition (Part II of II)**
> > > >
> > > > (Part II of II) We thank the reviewer for their comments. To re-state:
> > > >
> > > > ### 6th bullet point: Additional intuition/motivation for the experiments
> > > >
> > > > The crux of the paper and the experiments can be re-stated as follows: We know that there are inherent limitations to what can be achieved with distribution-free methods with finite sample: Conditional coverage is not possible, and the coverage guarantees assume exchangeability (i.e., no distribution shifts). Given these limitations, what approaches (and target quantities) should be used in practice with the deep networks? This is a question of immediate pressing concern given their deployment in real-world settings. More specifically, for classification:
> > > >
> > > > Point 1: (a) Can we nonetheless get close enough to conditional coverage to be generally useful in practice with deep networks, such as with localized conformal or Mondrian Conformal? (b) If not, what new approach can bridge the gap between the quantities produced by conformal predictors and those needed in practice?
> > > >
> > > > Point 2: Can we leverage the signals from KNN approximations of the deep networks for robustness to distribution-shifts?
> > > >
> > > > We note that another work (incidentally, by researchers with substantial previous work in CP) submitted after our work has also sought to address Point 1, but with a substantively different solution: Jin and Candès 2022: "Selection by Prediction with Conformal p-value" (https://arxiv.org/pdf/2210.01408.pdf); see in particular Section 1.1 (p. 3): **"a prediction set with marginal coverage guarantees is insufficient for selection...current tools for selection are all heuristic such as picking cases with a high predicted value and a relatively short prediction interval."**
> > > >
> > > > In our updated manuscript, we have emphasized evaluation of selection as it is a required minimal quantity in typical classification applications, and highlights a limitation deep learning researchers/practitioners should be aware of. By means of contrast with an example: With Conformal Prediction with a binary classifier, let's say we have seen 100,000 points with prediction sets of full cardinality (i.e., the sets contain {0,1}), and 10,000 points of cardinality 1. We might assume that the coverage of the points with cardinality 1 would be close to 1-alpha. However, that is an incorrect assumption: All we can say is that, over repeated experiments, the truth is contained in all 110,000 of the sets with a proportion of 1-alpha, a substantially weaker statement with a much narrower set of real-world use-cases.
> > > >
> > > > We show that the answer to Point 1.a. is also negative empirically within the standard approaches for localized conformal and Mondrian Conformal. That is, even if we partition the data with our strong signals from the KNNs, an additional mechanism beyond setting thresholds on the quantiles is needed. The subsequent work of Jin and Candès 2022 approaches this by running the Benjamini-Hochberg procedure on the conformal p-values to control FDR. We take a different approach. Given that we will typically want to partition the data with the signals from the metric learners as a guard on distribution shifts (Point 2 and see again Table 3), it becomes natural to instead re-cast the task to calibration via Venn Predictors. We show that the resulting quantity can be reliably obtained in very challenging settings. It is easy for practitioners to understand: It is simply the empirical probability of similar points defined by q, d, the predicted class, and the 1-alpha quantile bucket for each class.
> > > >
> > > > Novelty: **"Reliable selection under distribution shift, if not infeasible, may require more involved techniques"** (Jin and Candès 2022, p. 16). Remarkably, we have proposed and evaluated the first (to the best of our knowledge) such approach that is at least robust to changes in the data partitions (q and d). Given the wide-spread use of Transformers, coupled with concerns that large language models "do not know what they do[n't] know", our method and work is likely to be of wide, general interest.

---

### Author Response · Authors · 2022-11-18
**Revision**

We thank the reviewers for their comments and suggestions.

The present work is likely to be of general interest for the ICLR community given the strong empirical results, and principled theoretical justification. We present a general approach for uncertainty quantification for Transformer networks that works well even in very challenging settings (low-accuracy, class-imbalanced, distribution shifted). We are not aware of an alternative calibration approach that would have these characteristics. Our approach is particularly relevant given that Transformers are presently the core architecture for large language models currently being deployed in real-world settings.

Based on feedback from the reviewers, in our revision, we have re-cast the presentation in terms of selective classification, to reflect the typical use-case of the output. More specifically, in this view, the Mondrian Conformal Predictors simply serve as weak selective classifiers (i.e., the underlying predictive model), which are then calibrated via the Venn Predictor. We have made one slight modification to the previous algorithm which simplifies and tightens the theoretical argument/under-pinning: Instead of taking the average over the augmented distributions of the Venn Predictor (i.e., the empirical probability for a point when including the point itself, assuming a given label), we take the minimum value. In this way, the selective classifier is guaranteed to be well-calibrated in a conservative, but practically useful, sense. Additionally, to avoid complicating the theoretical argument, we simply create disjoint splits for the Mondrian Conformal Predictors and the Venn Predictors rather than using a leave-one-out approach. In practice, we do not observe a difference in the empirical results with the sample sizes of these tasks/data (either in overly reduced statistical efficiency---50% of the previous $D_\rm{ca}$ is still large enough for these tasks---nor in the other direction in terms of non-IID dependence when using leave-one-out), but it is simpler just to split  $D_\rm{ca}$ in two for the purposes here, and we leave further discussion of the leave-one-out approach to future work.

We consider the approach rather parsimonious overall, especially given the takeaway (reliable class-wise calibration, including over distribution shifts), but nonetheless, it does have a number of moving parts. We have added Figure 1 to provide an overview of how the different components interact.

We have added the ICLR Reproducibility statement and Ethics Statement before the references.

We have added justification for our heuristics (up-weighting via a KNN localizer and optionally restricting to the q=K partition), and we have provided additional details about training the KNNs in the Appendix.

We have aimed to make our paper as self-contained as possible, while also not inundating the reader with an unnecessarily long Appendix, and balancing the interests of distinct communities (ML, NLP, and Statistics).

To make the key results easier to follow, we now only focus on evaluation in terms of singleton sets/selective classification.

Unlike Conformal Predictors, the Venn-ADIMT selective classifiers are of immediate practical utility. Some example use-cases for reliable class-conditional selective classification include:

- Routing to reduce overall computation (e.g., use small, fast models, only deferring to larger models for rejected predictions)

- Higher-risk settings where less confident predictions must be sent to humans for further adjudication

- More generally, an unusual and advantageous aspect of the Venn-ADMIT Predictor, and which further distinguishes it from post-hoc Platt-scaling-style calibration, is a degree of inherent example-based interpretability: The calibrated distribution for a point is a simple transformation of the empirical probability among similar points, with partitions determined by a KNN that can be readily inspected. This matching component yields a direct avenue for addressing group-wise fairness: In principle, known group attributes can be incorporated as categories to ensure group-wise calibration.

*Note that the quantities produced by Conformal Predictors are not ideal for any of the aforementioned use-cases.* Or to put it another way, now that we have reliable selective classification, as presented here, the use-cases for the raw conformal prediction sets become more limited.

We now have a rather stronger handle on uncertainty over Transformers (at least in non-adversarial, non-concept-shift settings) than prior to this work. The costs are additional computation (space and time of the instance-based learners); coarser quantities than single probabilities; the need for labeled held-out sets of non-trivial size; and abstention from predicting over some points, but all of these costs are prospectively manageable in many real-world settings, especially given the avoided costs of unreliable predictions.

---

### Decision · Program_Chairs · 2023-01-20

**Decision:**

Reject

**Justification For Why Not Higher Score:**

N/A

**Justification For Why Not Lower Score:**

N/A

**Metareview: Summary, Strengths And Weaknesses:**

This paper considers the problem of approximate conditional coverage for prediction sets obtained by conformal prediction when the predictor is a transformer network. This is a very important and challenging problem in general.

The main idea behind the Venn-ADMIT method to solve this problem is very good, but the paper is not well-written and it is very hard to understand as echoed by all the reviewers'.
- I had a hard time understanding the meaning of "Definition 2". It needs better justification because every proposition about the method's robustness is directly defined in the context of well-calibrated selective classifiers. So I was not able to understand the theoretical implications of the approach.
- The method for creating weak and well-calibrated selective classifiers should have been introduced and explained earlier.
- The paper has good empirical results, but based on my understanding, theoretical guarantees are weak.

The fact that even people working in conformal prediction are not able to understand the details, the paper definitely needs to be re-written for clarity and precision. This is the main reason for my recommendation to reject the paper. I think the research direction and the method is promising, and I strongly encourage the authors' to carefully think about the exposition and writing for resubmission. One additional point to make clear is what aspects of the method are specific to transformers and what precludes from generalization to other deep networks.

**Summary Of Ac-Reviewer Meeting:**

N/A